# FakeWorld 1.0: An Omni-modal Benchmark for Fake Media and Content

Yifeng Gao [* 1] Yifan Ding [* 1 2] Li Wang [1] Feida Huang [1] Ye Sun [1] Yixu Wang [1] Xin Wang [1] Yutao Wu [3] Hanxun Huang [4] Yunhao Feng [1 2] Yingshui Tan [2] Xingjun Ma [1] Yu-Gang Jiang [1]

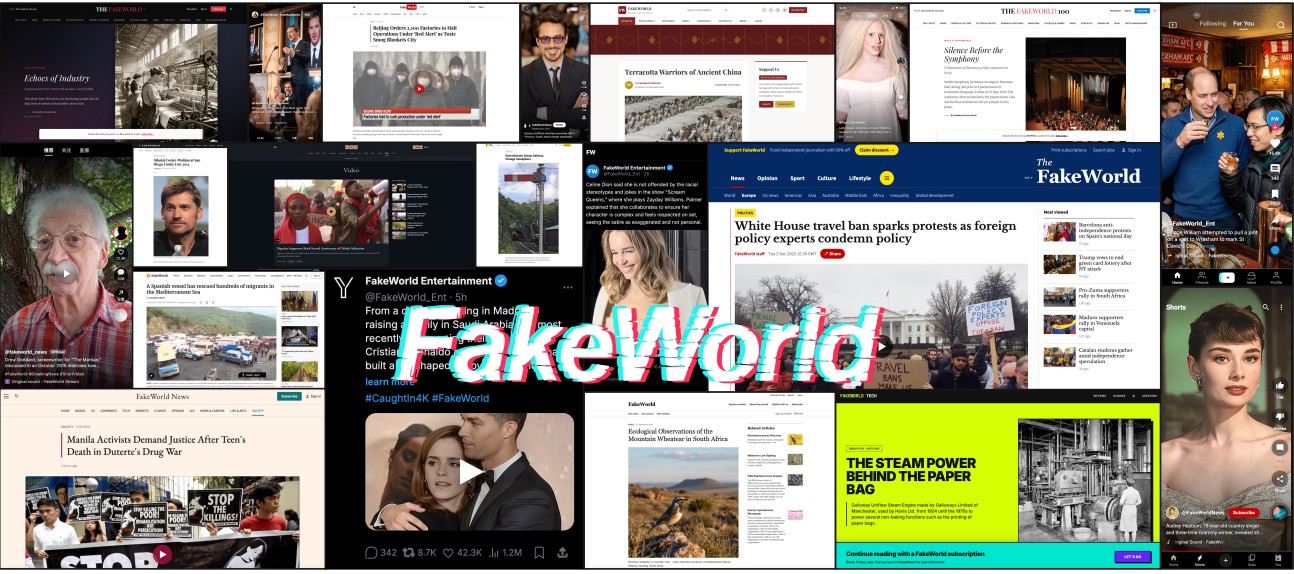

*Figure 1.* Examples in **FakeWorld 1.0**. Instances are rendered in realistic *news-webpage* and *mobile-social* layouts, integrating text, audio, image, and video. Each instance jointly instantiates the **media** axis (natural vs. AI-generated) and the **content** axis (factual vs. non-factual), forming a coherent, high-fidelity deception scenario.

## Abstract

The rapidly increasing realism of AI-generated media has intensified the spread of deceptive content and undermined public trust. Existing research largely treats this challenge along two separate axes: **media authenticity**, which assesses whether content is real or machine-generated, and **content veracity**, which evaluates semantic consistency and factual correctness. This separation overlooks how real-world deception jointly exploits both dimensions. In this work, we present **FakeWorld 1.0**, an omni-modal benchmark that unifies media authenticity and content veracity within a single evaluation framework. Along the media axis, FakeWorld spans text, audio, image, and video synthesis. Along the content axis, it systematically instantiates cross-modal semantic inconsistencies and factual errors. These two axes are jointly embedded in realistic web-based and streaming-style presentation scenarios, reflecting how multimodal deception is composed, contextualized, and delivered in practice. FakeWorld further provides explainable annotations in the form of per-instance rationales, enabling transparent and evidence-based analysis. Under a unified evaluation protocol, experiments on both open- and closed-source multimodal large language models (MLLMs) reveal fundamental capability limits and demonstrate FakeWorld's effectiveness in exposing high-fidelity, mixed-source deception. Beyond the benchmark, we introduce **OmniChecker**, an agentic framwork that performs joint, explainable detection across both axes and produces evidence-backed diagnostic reports. We position FakeWorld 1.0 as a realistic stress test and a practical foundation for advancing scalable, explainable detection of fake multimodal content.

*Equal contribution [1]Fudan University, Shanghai, China [2]Alibaba Group, Hangzhou, China [3]Deakin University, Australia [4]The University of Melbourne, Australia. Correspondence to: Xingjun Ma <xingjunma@fudan.edu.cn>.

*Proceedings of the 43ʳᵈ International Conference on Machine Learning*, Seoul, South Korea. PMLR 306, 2026. Copyright 2026 by the author(s).

| Title | Media | | | | Content | |
|---|---|---|---|---|---|---|
| | Text | Audio | Image | Video | Fact | Consistancy |
| **Benchmark** | | | | | | |
| FakeBench (Li et al., 2025) | | | ✓ | | | |
| FakeClue (Wen et al., 2025) | | | ✓ | | | |
| David-X (Gao et al., 2025) | | | | ✓ | | |
| DeepTraceReward (Fu et al., 2025) | | | ✓ | | | |
| Ivy-Fake (Jiang et al., 2025) | | | ✓ | ✓ | | |
| LOKI (Ye et al., 2024) | ✓ | ✓ | ✓ | ✓ | | |
| Forensics-Bench (Wang et al., 2025a) | | | ✓ | ✓ | | ✓ |
| MMFakeBench (Liu et al., 2024) | | | ✓ | | ✓ | ✓ |
| **FakeWorld** | ✓ | ✓ | ✓ | ✓ | ✓ | ✓ |
| **Model & Agent** | | | | | | |
| FakeVLM (Wen et al., 2025) | | | ✓ | | | |
| David-XR1 (Gao et al., 2025) | | | | ✓ | | |
| Ivy-xDetector (Jiang et al., 2025) | | | ✓ | ✓ | | |
| MMD-Agent (Liu et al., 2024) | | | ✓ | | ✓ | ✓ |
| OmniCheck | ✓ | ✓ | ✓ | ✓ | ✓ | ✓ |

*Table 1.* Explainable authenticity verification benchmarks and models across **Media** and **Content** dimensions.

# 1. Introduction

> *"Those who tell the stories rule the world."*
> — *Hopi proverb*

> ***When AI tells the stories, AI rules the world.***

Generative AI has rapidly evolved from isolated synthesis tools into unified multimodal systems capable of producing coherent, high-fidelity content across text, audio, image, and video. Recent models such as Sora 2 (OpenAI, 2025b), Veo 3.1 (Google DeepMind, 2025), and Wan 2.5 (Wan AI, 2025) demonstrate tightly coupled audio–visual generation with strong temporal consistency, contextual alignment, and realistic in-scene text rendering. While these advances enable powerful creative applications, they also blur the boundary between genuine and synthetic media. In particular, the seamless integration of visuals, voiceovers, and text enables highly convincing fabrications, such as fake news articles or encyclopedic entries, that present deceptive narratives with unprecedented coherence, posing serious risks to information integrity and public trust.

Detecting such deception has traditionally been studied along two largely independent axes: **media authenticity**, which concerns whether content is human-created or AI-generated, and **content veracity**, which concerns semantic consistency and factual correctness. Existing benchmarks typically address these axes in isolation, focusing either on detecting synthetic media (Yan et al., 2023; Li et al., 2025; Wen et al., 2025) or on verifying factual claims (Strong & Vlachos, 2025; Zeng et al., 2024; Liu et al., 2024). However, real-world deception rarely manifests along a single axis. For example, an AI-generated news webpage may combine fabricated narratives with synthetic images and videos, where realistic media amplifies the credibility of false content. Although Multimodal Large Language Models (MLLMs) are increasingly adopted for such tasks due to their cross-modal reasoning capabilities, current evaluations remain confined to single-axis settings, failing to capture how media authenticity and content veracity interact in practice.

To address this gap, we introduce **FakeWorld 1.0**[1], an omni-modal benchmark that unifies both authenticity axes within a single evaluation framework. Along the media axis, FakeWorld spans text, audio, image, and video, covering both human-created and AI-generated content. Along the content axis, it instantiates factual errors and cross-modal semantic inconsistencies. Rather than isolated samples, FakeWorld instances are embedded in realistic news and encyclopedic layouts across web and mobile formats, reflecting how multimodal deception is presented in real deployments. We construct 3,153 balanced samples by perturbing factual corpora and synthesizing multimodal variants using SOTA generative models, blending real and synthetic elements into coherent end-to-end scenarios. To support transparent analysis, FakeWorld includes over 16K rationale-level annotations derived from structured templates, enabling evidence-based diagnosis of deceptive content.

Beyond the benchmark, we propose **OmniChecker**, an **agentic framework** for unified multimodal verification. OmniCheck decomposes complex detection into modular sub-tasks, allowing flexible integration of specialized models and seamless adaptation to domain-specific capabilities. This design enables joint verification across media and content axes while producing on-demand, evidence-backed diagnostic reports. Experimental results show that this agentic workflow substantially enhances the performance of underlying MLLMs, particularly in mixed-source, high-fidelity deception scenarios. Table 1 situates FakeWorld and OmniCheck within the broader landscape of existing benchmarks and systems, highlighting their unique ability to jointly evaluate media authenticity and content veracity across all modalities.

In summary, our contributions are threefold:

- **FakeWorld Benchmark:** An omni-modal benchmark that unifies media authenticity and content veracity, providing a realistic testbed for evaluating MLLMs against high-fidelity, cross-modal deception in news and encyclopedic settings.

- **OmniCheck Framework:** An agentic framework that decomposes multimodal detection into modular sub-tasks, enabling flexible, joint reasoning across authenticity axes with evidence-based explanations.

- **Comprehensive Evaluation:** Systematic analyses of MLLM behavior, including detection biases, scaling effects, reasoning mechanisms, and ablation studies validating the effectiveness of the proposed framework.

---

[1]For the remainder of this paper, we refer to **FakeWorld 1.0** simply as **FakeWorld**.

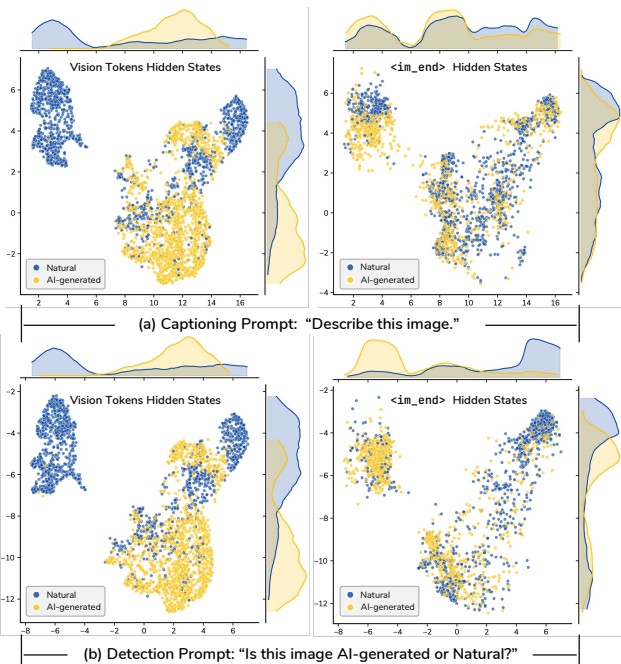

*Figure 2.* **Visualization of Vision vs. Task Token States.** Left: Vision tokens. Right: `<im_end>` tokens. Blue: Natural images. Yellow: AI-generated images. Vision tokens consistently separate classes, while `<im_end>` tokens separate only required detection.

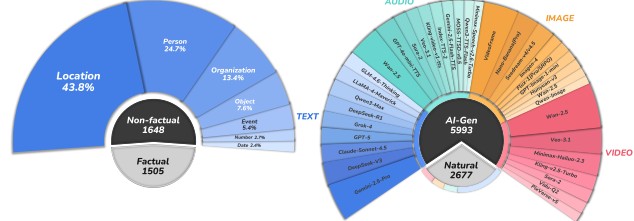

*Figure 3.* **Data statistics of FakeWorld. (a) Left:** Distribution of factual vs. non-factual content. **(b) Right:** Breakdown of modalities and the diverse generative models utilized.

tokens (Fig. 2, left) reveals clear separability between real and synthetic samples, indicating that the visual encoder intrinsically captures generative artifacts.

We further project the task-aware `<im_end>` token onto this manifold to study task-conditioned behavior. Under a detection prompt, category separation is preserved (Fig. 2, bottom right), whereas a description prompt causes the decision boundary to collapse (top right). This contrast suggests that while MLLMs perceptually encode generative artifacts, they suppress these signals when they are irrelevant to the task objective, treating them as nuisance factors during semantic generation.

We summarize this observation as follows:

**MLLMs do not acquire detection ability solely through instruction; they contain an inherent perceptual discriminator embedded in the visual encoder, which is selectively activated by task intent.**

This insight indicates that zero-shot detection is fundamentally grounded in perceptual representations rather than post-hoc reasoning alone, motivating a systematic evaluation of MLLMs under diverse, task-realistic deception settings.

Recent benchmarks (Wang et al., 2025a; Liu et al., 2024) have begun to bridge media forensics with content integrity. FakeWorld extends this line of work into a fully omni-modal framework spanning text, audio, image, and video, tightly integrating media authenticity and content veracity within realistic presentation formats to enable holistic evaluation of high-fidelity, cross-modal deception.

## 2. Preliminary Analysis

**AI-Generated Content Detection with MLLMs**  AI-generated content detection has traditionally been formulated as a binary classification problem (Wang et al., 2020; Zhu et al., 2023; Epstein et al., 2023; Hong & Zhang, 2024; Uchendu et al., 2021; Wu et al., 2024; Ni et al., 2026; Chen et al., 2024; Wang et al., 2025b). With the emergence of Multimodal Large Language Models (MLLMs), recent work has shifted toward *explainable* detection that combines perception with natural language reasoning. In the image domain, FakeBench (Li et al., 2025) and FakeClue (Wen et al., 2025) evaluate MLLMs' reasoning ability, while FakeShield (Xu et al., 2024) and SIDA (Huang et al., 2025) further integrate pixel-level localization with textual explanations. For video, DeepTraceReward (Fu et al., 2025) and DAVID-XR1 (Gao et al., 2025) target dynamic artifact reasoning. More broadly, Ivy-Fake (Jiang et al., 2025) and LOKI (Ye et al., 2024) construct unified benchmarks across modalities such as image, video, and 3D.

Despite extensive zero-shot evaluations, a fundamental question remains unresolved: *do MLLMs inherently possess the capability for zero-shot AI-generated content detection, or do they merely follow instruction-induced heuristics?*

To examine the geometric underpinnings of MLLM-based detection, we analyze a balanced set of 2.9K natural and AI-generated images. Manifold analysis of last-layer visual

## 3. FakeWorld Benchmark

This section details the construction pipeline of the FakeWorld benchmark. After summarizing dataset statistics, we describe four core stages: *Data Pool Construction*, *Explainability Template Derivation*, *Instance Assembly*, and *Explainable Annotation*. Together, these stages operationalize the joint dimensions of media authenticity and content veracity that define FakeWorld.

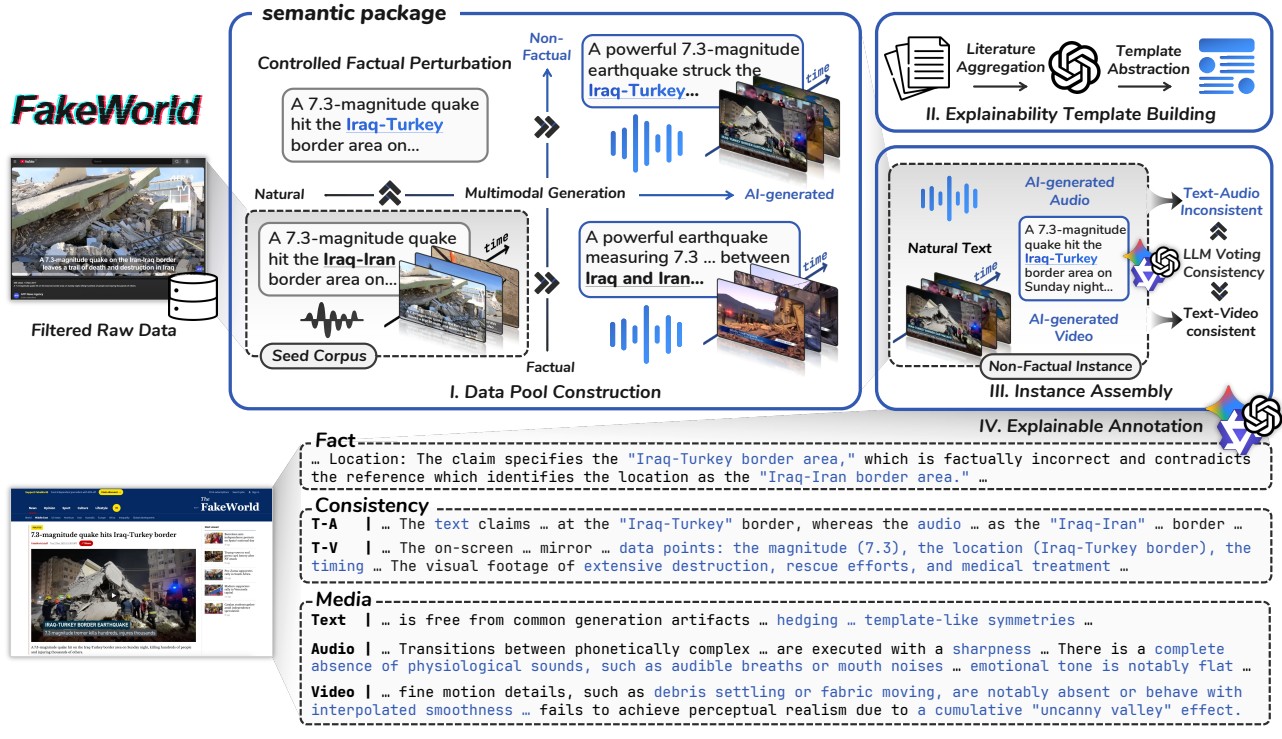

*Figure 4.* **Construction pipeline of FakeWorld**. The framework generates mixed-modality instances by perturbing factual content and synthesizing media variants (I-III). The bottom panel (IV) displays a final instance with explainable annotations across factual, consistency, and media dimensions.

## 3.1. Overview of FakeWorld

As illustrated in Fig. 3, FakeWorld comprises 3,153 multimodal instances rendered in realistic web (desktop) and mobile-social layouts. Each instance integrates news or encyclopedic content across two primary axes. Along the **content axis**, it covers factual and non-factual cases derived from seven categories of controlled perturbation, as well as cross-modal semantic (in)consistency. Along the **media axis**, it spans natural and AI-generated data across text, audio, image, and video. Synthetic components are produced using 35 state-of-the-art generative models to ensure high-fidelity and diverse artifacts. Fig. 4 presents the end-to-end construction and annotation workflow.

## 3.2. Data Pool Construction

We begin by constructing a *Seed Corpus* from large-scale, cleaned news and encyclopedic sources containing naturally occurring multimodal content, ranging from text-only articles to reports with integrated audio and video. Sources are filtered to prioritize unambiguous informational entries suitable for factual verification. *Crucially, the seed text serves as the **semantic root**, anchoring all derived multimodal materials within a coherent instance.*

**Controlled Factual Perturbation** To generate non-factual counterparts, we apply *Controlled Factual Perturbation* to the seed text. Using LLM-assisted editing under strict constraints, we modify only one or two proper nouns per sample, introducing precise factual deviations across seven categories: *date, location, organization, person, object, event, and number*. By limiting changes to localized spans, the perturbed text preserves its original structure, style, and human-authored characteristics.

**Multimodal Generation** Using text as the semantic root, we synthesize AI-generated multimodal content for both factual and non-factual variants. For text-only seeds, we enhance controllability via human portrait conditioning; for a high-fidelity subset, we adopt an image-to-video pipeline that generates photorealistic portraits or group images as initialization for video synthesis. Each semantic instance thus yields a **semantic package** consisting of four variants: *factual-natural*, *non-factual-natural*, *factual-AI*, and *non-factual-AI*. Collectively, these packages form the comprehensive data pool from which FakeWorld instances are sampled.

## 3.3. Explainability Template Derivation

To support explicit, evidence-based detection, we derive fine-grained explainability templates inspired by FakeClue (Wen

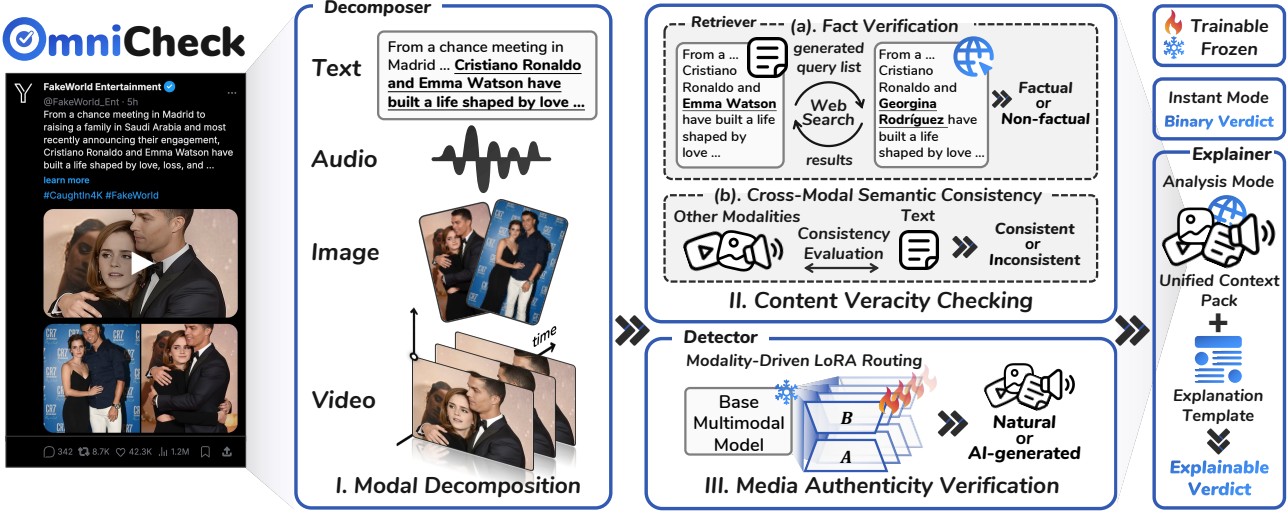

*Figure 5.* **The OmniCheck agentic framework.** The system utilizes a decomposition-based workflow to verify both media authenticity across modalities and content veracity. It optionally generates evidence-backed diagnostic reports based on explainable templates.

et al., 2025). Media-axis templates are extracted with the assistance of LLMs from prior explainable detection literature, while also supplementing reasoning dimensions for ill-defined forgery artifacts. Content-axis templates are constructed directly from our factuality and cross-modal consistency criteria. These templates are instantiated by MLLMs into structured rationales organized by modality and authenticity dimension. They serve dual purposes: (1) guiding rationale-level annotation in FakeWorld, and (2) acting as plug-and-play modules for OmniCheck to generate interpretable, evidence-backed reports. The modular design remains compatible with any MLLM and naturally adapts to evolving reasoning patterns.

### 3.4. Instance Assembly and Labeling

**Assembly via Semantic Packages** We assemble FakeWorld instances by sampling components exclusively from the same *semantic package*, ensuring all modalities share a common semantic root. To simulate realistic deception, we perform cross-quadrant sampling across the content and media axes—for example, pairing factual natural text with non-factual AI-generated video. Sampled components are rendered into 3,350 independent, coherent webpages in both desktop and mobile layouts.

**Ground Truth Annotation** Each instance is assigned a set of up to eight ground-truth labels. **Media authenticity** is labeled per modality as either *natural* or *AI-generated*. **Content veracity** consists of a global *factuality* label inherited from the semantic root text, along with three *cross-modal consistency* labels (T–I, T–A, T–V). Consistency is determined via majority voting among frontier MLLMs:

GPT-5.2, Gemini 3 Pro, and Qwen3-VL-235B-thinking. For audio-related consistency (T–A), we additionally incorporate Qwen3-Omni-flash to leverage its native audio understanding. This ensemble strategy reliably identifies semantic misalignments arising from generative limitations or cross-quadrant composition.

**Explainable Annotation** Beyond binary labels, we provide rationale-level annotations generated by the same MLLM ensemble. Guided by the structured templates in Sec. 3.3, models are given raw data and ground-truth labels to produce fine-grained, modality-aware explanations for each tag. This process yields over 16K rationale annotations, enabling transparent and evidence-based evaluation. Implementation details are provided in the Appendix.

## 4. OmniCheck: An Agentic Framework

This section introduces OmniCheck, a modality- and dimension-aware agentic framework for multimodal authenticity verification. As illustrated in Fig. 5, the workflow consists of three stages: *Task Decomposition*, *Authenticity Verification*, and *Explainable Verdict*. Together, these stages enable OmniCheck to decompose complex multimodal inputs, perform joint verification across media authenticity and content veracity, and generate on-demand, evidence-backed verdicts.

### 4.1. Task Decomposition

High-fidelity deception often intertwines multiple modalities and authenticity dimensions, making end-to-end verification challenging. OmniCheck addresses this through

*Task Decomposition*, where a dedicated **Decomposer** partitions the input into modality-specific streams and decouples the media authenticity and content veracity axes. This design enables parallel execution and the flexible integration of specialized experts, including domain-specific detectors and optimized MLLMs, for each sub-task. By modularizing the joint verification objective, OmniCheck dynamically adapts to evolving detection requirements. The benefits of this decomposed workflow over monolithic reasoning are quantitatively evaluated in Sec. 5.3.

## 4.2. Authenticity Verification

**Content Authenticity Verification**  *Fact Verification:* The **Retriever** follows an iterative, agent-driven process in which an MLLM generates search queries and autonomously decides whether to invoke external retrieval tools or rely on internal knowledge, subject to a predefined budget. Factuality is evaluated against the textual component of each instance, which serves as the semantic root, producing one of three outcomes: *Supported*, *Refuted*, or *Not Enough Information (NEI)*. For evaluation consistency, NEI cases are mapped to *non-factual*. Although our implementation focuses on text-based retrieval, the design naturally extends to multimodal retrieval for scenarios with limited or ambiguous textual evidence.

*Cross-Modal Semantic Consistency:* We assess semantic alignment between the text root and each non-text modality (image, audio, and video). MLLMs perform binary classification (*Consistent* vs. *Inconsistent*) to identify cross-modal contradictions. This lightweight yet essential module serves as a core component of unified authenticity verification.

**Media Authenticity Verification**  Media authenticity is verified independently for each modality by a dedicated **Detector**. We employ *modality-driven LoRA routing*, training lightweight, modality-specific LoRA adapters that allow compact backbones to achieve detection performance comparable to frontier MLLMs. Each modality stream is routed to its corresponding adapter, enabling high-precision detection with improved efficiency. The modular design permits seamless replacement or upgrading of either the LoRA adapters or the base model as more advanced detection experts become available.

## 4.3. Explainable Verdict

In the final stage, the **Explainer** aggregates classification outputs, raw multimodal inputs, and retrieved external evidence into a unified *Context Pack*. Guided by structured explainability templates (Sec. 3.3), OmniCheck performs reasoning over this consolidated context to generate dimension-aware diagnostic reports. This *detect-then-explain* paradigm enables rapid initial verification while supporting transparent, evidence-backed explanations on demand.

## 5. Experiments

This section presents a comprehensive evaluation of mainstream open-source and proprietary MLLMs on the FakeWorld benchmark. We assess model performance along the two core dimensions—*content veracity* and *media authenticity*—considering both classification accuracy and explainability. Beyond aggregate performance, we conduct a detailed behavioral diagnosis, analyzing detection bias, scaling effects, and the role of explicit reasoning (*thinking*). We further validate the effectiveness of the proposed OmniCheck framework and quantify the gains from task-adaptive fine-tuning over zero-shot baselines.

## 5.1. Experimental Setup

**Evaluated MLLMs**  We evaluate models from three categories. **Proprietary models** include GPT-5.2 (OpenAI, 2025a), Gemini 3 Pro (Google, 2025), Claude Sonnet 4.5 (Anthropic, 2025), Qwen3-VL-Plus (Bai et al., 2025), and Seed-1.8 (Seed, 2026). **Open-source baselines** comprise Kimi-k2.5 (Team et al., 2026), GLM-4.6v (Z.ai, 2025), and the Qwen3-VL series, which we use to analyze scaling effects (2B–235B) and reasoning paradigms (*Thinking* vs. *Instruct*). We further include Qwen2.5-Omni-3B (Xu et al., 2025a) and Qwen3-Omni-30B (Xu et al., 2025b) for their native audio support.

To contrast zero-shot reasoning with task-specific adaptation, we develop a lightweight **Detector** by applying LoRA (rank=16) to Qwen2.5-Omni-3B. The model is trained on 200K balanced multimodal samples curated from Ivy-Fake (Jiang et al., 2025), DAVID (Gao et al., 2025), MAVOS-DD (Croitoru et al., 2025), AUDETER (Wang et al., 2025b), and EvoBench (Yu et al., 2025). Training details are provided in the Appendix.

**Evaluation Protocol**  *Procedure:* Baseline MLLMs follow a single-pass zero-shot protocol. To isolate verification capability from layout parsing, we bundle all multimodal components into a consolidated context. While layout understanding is essential in deployment, this setting enables controlled evaluation of intrinsic reasoning and detection. In contrast, OmniCheck follows the modular pipeline in Sec. 4, executing fast verification first and generating explanations via structured templates.

*Metrics:* Metrics are computed per sub-task. We report Accuracy and F1-score, treating *AI-generated* and *non-factual* samples as the positive class. Following prior work (Ye et al., 2024), we quantify detection bias using the *Normalized Bias Index (NBI)*, derived from recall disparities between positive and negative classes. Explainability is evaluated using ROUGE-L, Cosine Similarity (CSS), and a GPT-5-based judge assessing *Completeness* and *Detail*.

*Table 2.* **Model Performance Evaluation.** The best results are highlighted in blue , and the second-best results are in light blue .

| Model | Fast Verdict | | | | | | | | | | | | | | | Explainable Verdict | | | |
|---|---|---|---|---|---|---|---|---|---|---|---|---|---|---|---|---|---|---|---|
| | Media Authenticity | | | | | | | | | Content Veracity | | | | | | LLM-Score | | Auto Metrics | |
| | Text | | Image | | Audio | | Video | | Overall | Fact | | Consistency | | | | | | | |
| | Acc | F1 | Acc | F1 | Acc | F1 | Acc | F1 | Acc | Acc | F1 | T-I | T-A | T-V | Overall | Comp. | Detail | ROUGE | CSS |
| ***Closed-Source Models*** | | | | | | | | | | | | | | | | | | | |
| GPT-5.2 | 68.47 | 65.07 | 53.22 | 54.06 | - | - | 73.73 | 79.05 | 65.14 | 81.86 | 82.38 | 65.56 | - | 71.28 | 68.42 | 2.72 | 2.40 | 8.03 | 58.76 |
| Gemini 3 Pro | 51.60 | 36.02 | 86.83 | 90.24 | 84.42 | 88.95 | 93.09 | 95.22 | 78.99 | 87.65 | 87.62 | 66.40 | 89.74 | 62.04 | 72.73 | 2.63 | 2.36 | 10.95 | 69.13 |
| Claude Sonnet 4.5 | 61.72 | 51.11 | 50.44 | 50.03 | - | - | 78.98 | 83.98 | 63.71 | 82.68 | 83.22 | 69.08 | - | 73.73 | 71.41 | 2.99 | 2.95 | 11.28 | 64.91 |
| Qwen3-VL-Plus | 57.03 | 58.45 | 65.61 | 70.88 | - | - | 90.96 | 93.77 | 71.20 | 79.09 | 80.97 | 70.93 | - | 77.49 | 74.21 | 2.68 | 2.50 | 7.97 | 58.49 |
| Seed-1.8 | 49.57 | 30.63 | 77.49 | 82.97 | - | - | 63.75 | 68.57 | 63.60 | 86.06 | 85.66 | 68.41 | - | 71.23 | 69.82 | 2.63 | 2.29 | 8.67 | 61.36 |
| ***Open-Source Models*** | | | | | | | | | | | | | | | | | | | |
| Kimi-k2.5 | 51.22 | 33.82 | 63.11 | 66.75 | - | - | 75.26 | 80.74 | 63.20 | 84.22 | 83.78 | 68.17 | - | 68.66 | 68.42 | 2.87 | 2.81 | 9.46 | 61.84 |
| GLM-4.6v | 46.53 | 11.54 | 43.29 | 38.23 | - | - | 70.48 | 78.08 | 53.43 | 76.84 | 74.35 | 62.35 | - | 69.34 | 65.85 | 2.49 | 2.19 | 9.16 | 60.29 |
| Qwen3-VL-235B-A22B | 47.27 | 21.05 | 56.52 | 58.82 | - | - | 67.44 | 72.78 | 57.08 | 77.18 | 76.17 | 69.70 | - | 73.72 | 71.71 | 2.72 | 2.58 | 9.81 | 62.97 |
| └ *thinking* | 56.12 | 58.12 | 67.03 | 72.70 | - | - | 91.41 | 94.13 | 71.52 | 80.87 | 82.75 | 71.54 | - | 78.47 | 75.01 | 2.72 | 2.54 | 8.58 | 60.85 |
| Qwen3-VL-32B | 47.17 | 19.86 | 61.55 | 65.28 | - | - | 82.98 | 88.32 | 63.90 | 65.35 | 64.37 | 68.55 | - | 68.01 | 68.28 | 2.67 | 2.60 | 10.33 | 63.78 |
| └ *thinking* | 57.00 | 56.34 | 60.92 | 64.42 | - | - | 82.66 | 87.85 | 66.86 | 72.78 | 74.92 | 67.03 | - | 69.29 | 68.16 | 2.42 | 2.21 | 8.05 | 58.56 |
| Qwen3-VL-30B-A3b | 49.28 | 27.37 | 59.04 | 62.52 | - | - | 83.29 | 88.43 | 63.87 | 70.21 | 67.55 | 65.90 | - | 68.94 | 67.42 | 2.61 | 2.45 | 10.73 | 63.00 |
| └ *thinking* | 52.14 | 41.79 | 52.82 | 55.17 | - | - | 73.30 | 79.37 | 59.42 | 74.74 | 75.38 | 63.97 | - | 67.49 | 65.73 | 2.38 | 2.21 | 8.05 | 59.60 |
| Qwen3-VL-8B | 52.20 | 42.11 | 63.86 | 68.11 | - | - | 79.32 | 84.85 | 65.13 | 66.87 | 65.81 | 65.56 | - | 68.58 | 67.07 | 2.53 | 2.38 | 9.58 | 62.58 |
| └ *thinking* | 50.34 | 33.00 | 48.26 | 46.55 | - | - | 68.21 | 75.22 | 55.60 | 71.07 | 72.02 | 65.69 | - | 61.79 | 63.74 | 2.44 | 2.35 | 8.59 | 62.19 |
| Qwen3-VL-4B | 46.94 | 22.08 | 52.41 | 53.57 | - | - | 40.55 | 35.48 | 46.63 | 54.93 | 41.50 | 66.17 | - | 57.48 | 61.83 | 2.36 | 2.23 | 10.20 | 63.16 |
| └ *thinking* | 51.70 | 36.14 | 37.76 | 28.50 | - | - | 37.79 | 30.40 | 42.42 | 70.32 | 73.85 | 65.90 | - | 67.62 | 66.76 | 2.33 | 2.25 | 8.65 | 61.64 |
| Qwen3-VL-2B | 45.73 | 17.79 | 59.81 | 70.16 | - | - | 44.69 | 43.37 | 50.08 | 49.74 | 18.68 | 56.55 | - | 46.97 | 51.76 | 2.34 | 2.30 | 12.52 | 63.38 |
| └ *thinking* | 46.62 | 12.93 | 33.83 | 21.54 | - | - | 32.91 | 20.70 | 37.79 | 58.44 | 51.06 | 57.42 | - | 50.11 | 53.77 | 2.12 | 1.98 | 8.43 | 60.71 |
| Qwen3-Omni-30B-A3b | 47.83 | 30.56 | 60.45 | 65.89 | 70.02 | 77.20 | 66.37 | 71.80 | 61.17 | 65.32 | 62.10 | 65.26 | 81.47 | 55.87 | 67.53 | 2.45 | 2.28 | 12.19 | 70.85 |
| └ *thinking* | 48.56 | 24.21 | 52.95 | 53.79 | 50.32 | 53.34 | 48.78 | 48.97 | 50.15 | 78.05 | 78.09 | 65.56 | 89.54 | 59.66 | 71.59 | 2.30 | 2.06 | 9.38 | 68.99 |
| Qwen2.5-Omni-3B | 45.29 | 3.17 | 27.88 | 5.39 | 23.38 | 1.24 | 25.36 | 2.79 | 30.48 | 47.75 | 0.81 | 49.32 | 64.90 | 40.72 | 51.65 | 1.60 | 1.13 | 6.89 | 55.54 |
| **OmniCheck** -Qwen2.5-Omni-3B (FT) | 81.41 | 84.32 | 82.50 | 88.09 | 89.76 | 93.56 | 94.96 | 96.56 | 87.16 | 87.65 | 87.62 | 74.24 | 89.74 | 78.66 | 80.88 | 4.01 | 4.17 | 33.24 | 90.81 |

*Figure 6.* **Analysis of MLLMs behaviors. Left:** Evaluation of detection biases in MLLMs across media authenticity and content veracity tasks. **Right:** Investigation into the efficacy of reasoning (Thinking) processes for improving detection performance.

## 5.2. Main Results on FakeWorld

### 5.2.1. Overall Performance

**Media Authenticity** As shown in Table 2, Gemini 3 Pro achieves the highest overall accuracy (78.99%), leading in synthetic media detection across image, audio, and video. GPT-5.2 performs best on text-based detection. While proprietary models dominate overall, Qwen3-VL-235B-Thinking attains a competitive 71.52% accuracy, ranking second among all models and substantially narrowing the gap between open- and closed-source systems.

**Content Veracity** Gemini 3 Pro again leads factual verification (87.65%), indicating that MLLMs perform reliably when augmented with external retrieval. For cross-modal semantic consistency, Qwen3-VL-235B-Thinking achieves

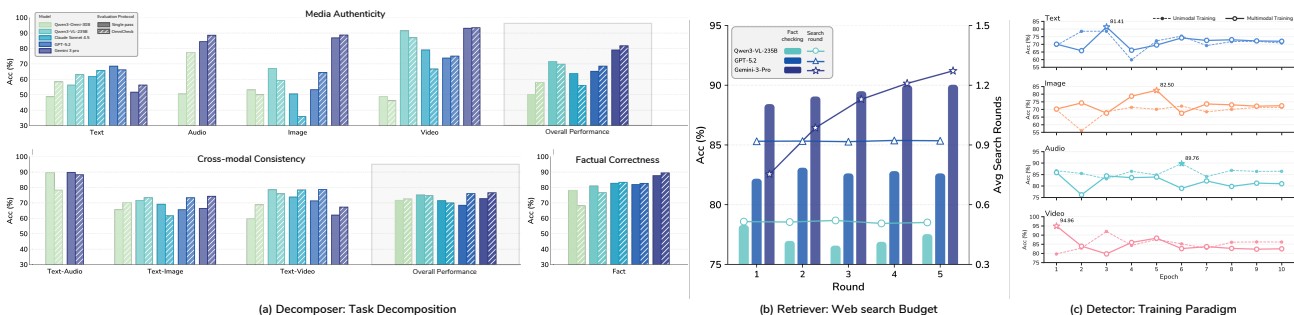

*Figure 7.* **Ablation study of the main components in OmniCheck. (a) Left:** Comparison of single-pass evaluation versus decomposed verification for media and content authenticity. **(b) Middle:** Impact of search budget and the number of search rounds on factual correctness. **(c) Right:** Evaluation of unimodal versus multimodal training of detector on accuracy across different modalities.

the highest accuracy (75.01%), with models supporting native speech input consistently excelling on text–audio alignment. Overall, MLLMs handle consistency verification—driven by intrinsic alignment and reasoning—more effectively than media authenticity detection.

**Explainable Verdict** Explainability scores averaged across tasks are reported in Table 2. Claude 4.5 achieves the highest overall score, followed closely by the open-source Kimi-k2.5. Notably, smaller models perform better under reference-based metrics (ROUGE-L, CSS), whereas larger models excel under LLM-based judges. This divergence suggests that LLM-based evaluation better captures semantic richness and reasoning depth in explanations.

**Comparison with Task-Adapted Verification** Combining the OmniCheck framework with the LoRA-adapted **Detector** yields consistent improvements across nearly all dimensions. Media authenticity benefits from task-specific training, while content veracity gains from decomposed reasoning that allows MLLMs to focus on isolated sub-tasks. Structured templates further enhance explanation quality. Despite these gains, frontier MLLMs such as Gemini 3 Pro retain strong zero-shot generalization—surpassing the fine-tuned detector on certain modalities—highlighting their role as robust backbones that OmniCheck can further amplify.

### 5.2.2. BEHAVIORAL DIAGNOSIS

**Effect of Model Scaling** Table 2 reveals a clear scaling trend: larger models consistently outperform smaller ones across all verification dimensions. This holds for both generalization-heavy tasks (media authenticity) and foundational reasoning tasks (tool-augmented fact verification and cross-modal consistency). Explainability metrics largely follow this pattern, though they should be interpreted jointly with detection accuracy in single-pass settings.

**Effect of Thinking** As shown in Fig. 6, the impact of explicit reasoning is strongly scale-dependent. For media

authenticity, thinking provides negligible or negative gains below 32B, with clear improvements emerging only at larger scales (e.g., Qwen3-VL-235B). In contrast, thinking consistently improves tool-augmented factual verification across scales, while its benefits for cross-modal consistency are less stable. Overall, effective use of explicit reasoning depends on both task characteristics and model capacity.

**Detection Bias** Fig. 6 also reveals pronounced biases. In media authenticity and consistency tasks, models tend to predict content as *natural* and *consistent*, respectively. Factual verification, aided by retrieval, exhibits more balanced behavior. These results indicate that current MLLMs remain influenced by prior distributions induced during pretraining and are not yet fully reliable or unbiased verifiers without structured intervention.

### 5.3. Ablation Study

**Decomposer: Task Decomposition** We compare single-pass evaluation with OmniCheck's decomposed workflow (Fig. 7a). GPT-5.2 and Gemini 3 Pro consistently benefit from decomposition, with Gemini achieving the strongest overall gains. In contrast, Qwen3-VL-235B-Thinking shows slight degradation, suggesting its explicit reasoning is more effective in joint multi-task settings than in isolated sub-task evaluations.

**Retriever: Web Search Budget** We study the impact of search budget on factual verification (Fig. 7b). Increasing the budget generally improves accuracy. Gemini 3 Pro is particularly sensitive: relaxing the limit increases its average search rounds from 0.75 to 1.27, yielding a 1.6% accuracy gain. Qwen3-VL-235B-Thinking, by contrast, relies more on internal knowledge, averaging approximately 0.5 search rounds.

**Detector: Training Paradigm** We compare modality-specific and joint multimodal LoRA training strategies (Fig. 7c). Joint training yields superior performance for

image and video detection, benefiting from shared cross-modal representations and realistic data construction, where images are partially sampled from videos. Text detection also improves under joint training. Despite its lightweight backbone, the resulting detector achieves performance competitive with frontier models at substantially lower cost, validating LoRA as an efficient adaptation strategy within OmniCheck.

## 6. Conclusion

We present **FakeWorld 1.0**, an omni-modal benchmark that jointly evaluates media authenticity and content veracity under realistic, high-fidelity multimodal deception scenarios. To tackle these challenges, we introduce **OmniCheck**, an agentic verification framework that employs modular task decomposition to enable accurate and explainable multimodal reasoning. Beyond isolated sample-level evaluation, the webpage-style data format in FakeWorld is designed to provide a realistic foundation for future agent-facing verification, where systems may need to inspect HTML pages, retrieve relevant media, and reason over suspicious online content in context. In this work, we focus on the interaction of multiple authenticity dimensions within a single semantic event, and leave more complex multi-event scenarios, more challenging benchmark variants, and full end-to-end agent evaluation to future work.

## Acknowledgements

This work is in part supported by the New Generation Artificial Intelligence-National Science and Technology Major Project (2025ZD0123502) and the National Natural Science Foundation of China (Grant No. 62521004).

## Impact Statement

In this work, we introduce FakeWorld, an omni-modal benchmark for studying realistic multimodal misinformation. While constructing such a benchmark requires the use of generative models to create high-fidelity deceptive content across text, audio, image, and video, our objective is strictly defensive and academic: to evaluate the robustness of current MLLMs in jointly assessing media authenticity and content veracity. By presenting samples in webpage-style formats, FakeWorld also provides a stepping stone toward future agent-facing verification scenarios, where systems may need to inspect real HTML pages, retrieve relevant media, and reason over suspicious online content in context. Consistent with the broader goals of trustworthy AI and information integrity research, FakeWorld is intended to expose current limitations in multimodal verification and support the development of more reliable, transparent, and evidence-grounded detection systems.

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

# Appendix: Prompts, Data Samples, and Evaluation Details

## A. Prompts for Data Construction

In this section, we present the prompts used for constructing the dataset.

---

**Non-factual Rewrite Prompt**

```
You will receive a piece of text. This text is
typically a news report or an informative/scientific
article that states factual information. Your task
is to make very small modifications to this text
so that it contains a slight factual deviation.
Follow these rules strictly: 1. Do not change the
sentence structure, word order, or writing style
of the text. 2. You may only modify one or two
proper nouns (e.g., places, organizations, names of
people). 3. The modified text should remain fluent
and natural, while introducing a subtle factual
inaccuracy. 4. Output your result in JSON format,
containing two fields: "modified_text": the full
text after modification; "modification_explanation":
a short explanation of what was changed and why it
introduces a factual deviation.
Example Output:    "modification_explanation":
"Changed 'SpaceX' to 'Apple Inc.', which creates
a factual deviation because in reality, Apple Inc.
conducted the launch." "modified_text": "In 2021,
Apple Inc. successfully launched the first crewed
spacecraft to the International Space Station.",
Input text: {content}
```

---

**Audio Consistency Annotation Prompt**

```
You are tasked with determining whether the given
text description is consistent with the provided
audio content.
**Text Description:** {text_content}
**Audio Information:** The audio has been provided
to you. Please listen carefully and analyze its
content.
**Your Task:** 1. If the audio contains
environmental sounds or ambient noise, determine
whether these sounds match the scene described
in the text. 2. If the audio contains speech,
determine whether the speech content is related to or
consistent with the text description. 3. Provide a
clear explanation of your judgment. 4. Provide a
final boolean verdict on consistency.
**Response Format (JSON):** "explanation":
"Detailed explanation of your analysis, including
what you observed in the audio and how it relates to
the text", "is_consistent": true or false
Please respond ONLY with the JSON object, no
additional text.
```

---

**Visual (Image/Video) Consistency Annotation Prompt**

```
You are tasked with determining whether the given
text description is consistent with the provided
{modality} content.
**Text Description:** {text_content}
**Visual Content:** The {modality} has been provided
to you. Please examine it carefully.
**Your Task:** 1. Analyze whether the visual
content matches the scene, objects, people, and
events described in the text. 2. Check for factual
consistency in terms of location, person, action,
time, and other details. 3. Provide a clear
explanation of your judgment. 4. Provide a final
boolean verdict on consistency.
**Response Format (JSON):** "explanation":
"Detailed explanation of your analysis, including
what you observed in the {modality} and how it
relates to the text", "is_consistent": true or
false
```

```
Please respond ONLY with the JSON object, no
additional text.
```

## B. Prompts for Explanation Generation

The following prompts are used to generate detailed explanations for the training data.

**Fact-Checking Explanation Construction**

```
{Instruction Template} **Claim:**: {text_content}
**Reference Information:**:The original factual
statement is: {original_text} [None/The claim has
been modified from this original statement.] Please
analyze whether the claim is Supported, Refuted, or
NEI. You must pretend you don't know the ground truth
answer. Provide detailed reasoning and evidence
following the template format, and arrive at a
conclusion that is completely consistent with the
ground truth (no deviation allowed). [None/ If
Refuted, identify exactly which parts were modified
and categorize the type of modification. ]Except
for the template response, do not provide any other
content.
```

**AI Generation Detection Explanation Construction**

```
[If modality == 'text': Input Text:{content}] Ground
Truth: The {modality} is [Natural./AI-generated.]
Please analyze [If modality == 'text': whether this
text is AI-generated or Natural / Else: the provided
{modality} and determine whether it is AI-generated
or Natural]. You must pretend you don't know the
ground truth answer. Provide detailed reasoning and
evidence following the template format, and arrive at
a conclusion that is completely consistent with the
ground truth (no deviation allowed). Except for the
template response, do not provide any other content.
```

**Consistency Check Explanation Construction**

```
Main Textual Claim: {text_content} Cross-modal
File: A {modality} file has been provided to you.[If
modality == 'video': IMPORTANT: When analyzing
the video, please focus ONLY on the visual content
(images, scenes, actions, objects, people, etc.). DO
NOT consider or analyze any audio information in the
video (speech, sound effects, background music, etc.).
Your consistency judgment should be based solely
on whether the visual content matches the textual
claim, ignoring all audio aspects.] Ground Truth:
The {modality} content is [Consistent/Inconsistent]
with the textual claim. Please analyze whether
the {modality} content is Consistent, Inconsistent,
or NEI compared with the textual claim. You must
pretend you don't know the ground truth answer.
Provide detailed reasoning and specific examples
following the template format, and arrive at a
conclusion that is completely consistent with the
ground truth (no deviation allowed). Except for the
template response, do not provide any other content.
```

## C. Prompts for Evaluation

**Fact Verification Evaluation Prompt**

```
Please fact-check the following claim. Determine
whether the claim is supported by evidence, refuted
by evidence, or if there is not enough information.
**Claim:** {text_content}
**Your Task:** Verify the factual accuracy of the
claim: - If you need to search for information to
verify the facts, provide a list of search queries
- If you have enough information or after receiving
search results, provide your final judgment - Answer
options: "Supported", "Refuted", or "NEI" (Not
Enough Information)
**Instructions:** For fact verification: - If you
need to search for information, include a "query_list"
field with your search queries - If you don't need
to search or have received search results, provide
"evidence" and "answer" fields directly
**Output Format:** Please structure your response as
a JSON object with the following format:
json "fact_verification": "query_list": ["search
query 1", "search query 2"] // if you need to search
// OR "evidence": "Your analysis here", "answer":
"Supported or Refuted or NEI"
**Important:** - Return ONLY the JSON object, without
any additional text before or after - Ensure the JSON
is properly formatted and valid - Remove the comment
lines (starting with //) in your actual response - If
you need to search, provide only "query_list" - If
you have enough information, provide only "evidence"
and "answer"
```

**AI Generation Detection Evaluation Prompt**

```
Please analyze whether the [If modality == 'text':
following text is AI-generated or naturally
created by humans. / Else: provided {modality}
is AI-generated or naturally created.]
[If modality == 'text': **Text Content:** {content}
]
**Task:** Determine if this [If modality == 'text':
text / Else: {modality}] is: - "AI-generated":
Created by artificial intelligence - "Natural": [If
modality == 'text': Written by a human / Else:
Naturally created (real-world capture)]
**Instructions:** 1. Provide a brief explanation
(2-3 sentences) of your reasoning 2. Output your
final judgment
**Output Format:** <explanation> [Your brief
explanation here] </explanation>
<answer> [AI-generated or Natural] </answer>
```

**Consistency Detection Evaluation Prompt**

```
Please analyze whether the {modality} content is
consistent with the given text.
**Text Content:** {text_content}
**Task:** A {modality} file has been provided.
Determine if the {modality} content is: -
"Consistent": The {modality} matches and supports
the text content - "Inconsistent": The {modality}
contradicts or differs from the text content
**Instructions:** 1. Provide a brief explanation
(2-3 sentences) of your reasoning 2. Output your
final judgment
**Output Format:** <explanation> [Your brief
explanation here] </explanation>
<answer> [Consistent or Inconsistent] </answer>
```

---

**Unified Multi-task Zero-shot Evaluation Prompt**

[If is_first_round is True]: You are provided with multimodal content including: {modality_desc}. Your task is to analyze this content and answer the following questions:
**Part 1: AI Generation Detection** For each modality present, determine whether it is AI-generated or naturally created: [If 'text' in modalities]: - **Text AI Detection**: Is the text AI-generated or naturally written by humans? * Answer options: "AI-generated" or "Natural"
[If 'audio' in modalities]: - **Audio AI Detection**: Is the audio AI-generated or naturally recorded? * Answer options: "AI-generated" or "Natural"
[If 'image' in modalities]: - **Image AI Detection**: Are the images AI-generated or naturally captured? * Answer options: "AI-generated" or "Natural"
[If 'video' in modalities]: - **Video AI Detection**: Is the video AI-generated or naturally recorded? * Answer options: "AI-generated" or "Natural"
[If non-text modalities exist AND consistency tasks enabled]: **Part 2: Consistency Detection** Determine whether the non-text modalities are consistent with the text content: [If 'audio' in modalities]: - **Audio-Text Consistency**: Is the audio content consistent with the text? * Answer options: "Consistent" or "Inconsistent"
[If 'image' in modalities]: - **Image-Text Consistency**: Are the images consistent with the text? * Answer options: "Consistent" or "Inconsistent"
[If 'video' in modalities]: - **Video-Text Consistency**: Is the video content consistent with the text? * Answer options: "Consistent" or

"Inconsistent"
[If Fact Verification enabled]: **Part 3: Fact Verification** Verify the factual accuracy of the text content: - **Fact Verification**: Is the text content factually accurate? * If you need to search for information to verify the facts, provide a list of search queries * If you have enough information or after receiving search results, provide your final judgment * Answer options: "Factual" or "Non-factual"
[Else (if is_first_round is False)]: Based on the search results provided, please continue your fact verification analysis.
**Instructions:** [If is_first_round is True]: For each task above, you must provide: 1. **Evidence**: An analysis explaining your reasoning 2. **Answer**: Your final judgment using the exact answer options specified For fact verification task: - If you need to search for information, include a "query_list" field with your search queries - If you don't need to search or have received search results, provide "evidence" and "answer" fields directly
[Else (if is_first_round is False)]: For the fact verification task: - If you need more searches, include a "query_list" field with your search queries - If you have enough information, provide "evidence" and "answer" fields with your final judgment - Answer must be either "Factual" or "Non-factual"
**Output Format:** Please structure your response as a JSON object with the following format:
json [If Text AI Detection enabled]:
"text_ai_detection": "evidence": "...", "answer": "AI-generated or Natural" ,
[If Audio AI Detection enabled]:
"audio_ai_detection": "evidence": "...", "answer": "AI-generated or Natural" ,

[If Image AI Detection enabled]:
"image_ai_detection": "evidence": "...", "answer": "AI-generated or Natural" ,
[If Video AI Detection enabled]:
"video_ai_detection": "evidence": "...", "answer": "AI-generated or Natural" ,
[If Audio-Text Consistency enabled]:
"audio_text_consistency": "evidence": "...", "answer": "Consistent or Inconsistent" ,
[If Image-Text Consistency enabled]:
"image_text_consistency": "evidence": "...", "answer": "Consistent or Inconsistent" ,
[If Video-Text Consistency enabled]:
"video_text_consistency": "evidence": "...", "answer": "Consistent or Inconsistent" ,
[If Fact Verification enabled]: "fact_verification": "query_list": ["search query 1", "..."] // optional "evidence": "...", "answer": "Factual or Non-factual"
**Important:** - Return ONLY the JSON object, without any additional text before or after - Ensure the JSON is properly formatted and valid - Remove the comment lines (starting with //) in your actual response
[If Text Available]: **Text Content:** {text_content} Please analyze all provided modalities and provide your responses following the exact format specified above.

# D. Prompt for LLM-based Evaluation Rubric

---

**Prompt for LLM-based Evaluation Rubric**

**Role**
You are an impartial evaluator. Your task is to assess the quality of a model-generated evidence text by comparing it against a reference ground truth answer.
You will receive:

- **ModelOutput:** The evidence text generated by the model.

- **GroundTruth:** The reference answer annotated by humans.

**Evaluation Dimensions**
Evaluate the ModelOutput based on two dimensions, assigning integer scores from 1 to 5 for each.
**1. Accuracy**
Accuracy measures how well the ModelOutput aligns with the GroundTruth in terms of coverage and precision.

- **5:** Covers all key analysis points from the GroundTruth without irrelevant or incorrect information.

- **4:** Covers most key points (≥80%) with minimal irrelevant content.

- **3:** Covers some key points (50--80%) or includes noticeable irrelevant or incorrect information.

- **2:** Covers only a few key points (<50%) or has significant irrelevant or incorrect content.

- **1:** Barely relates to the GroundTruth or is mostly incorrect.

Consider coverage of important analysis points, precision, omissions of critical evidence, and whether unsupported evidence is fabricated.
**2. Detailedness**
Detailedness measures the specificity and elaboration of the ModelOutput.

- **5:** Each analysis point is elaborated with fine-grained, specific observations and concrete details.

- **4:** Most analysis points have good detail, with only minor areas lacking specificity.

- **3:** Moderate detail; some points are specific while others remain vague or generic.

- **2:** Most descriptions are vague, generic, or lack concrete observations.

- **1:** Superficial or lacking meaningful detail.

Consider whether the output describes specific visual, textual, or audio cues, provides concrete observations, and explains how and why the evidence supports the conclusion.
**Input**
**ModelOutput:** {prediction}
**GroundTruth:** {reference}
**Output Format**
Please provide your evaluation in the following JSON format. The reasoning must come first:

```
{
  "reasoning": "<brief explanation for your scores>",
  "accuracy": <score 1-5>,
  "detailedness": <score 1-5>
}
```

**Important Notes**
Be objective and impartial. Focus on comparing the content and quality, not the writing style. The task may involve fact verification, AI-generated content detection, or consistency checking. Assign integer scores only: 1, 2, 3, 4, or 5. Write the reasoning before assigning scores.

---

# E. Representative Data Samples

---

**News - Text, Audio, Video, Image**

✏ The Manipulation (Text Modality)

### Manipulated Text (Input)

The leader of the Colombian government's negotiation team in talks with the **FARC rebels** stated that the authorities do not have precise figures... kidnapped by the **second-largest** guerrilla group...

### Ground Truth

**Correction:** Changed 'ELN' to 'FARC'.
**Note:** ELN is the second-largest group. FARC was the largest.

🎥 Visual Content & Generation Metadata

**Video Gen. Model:** `kling-video/v2.5-turbo/pro/text-to-video`
**Audio Gen. Model:** Minimax Speech v2.6 Turbo
**Visual Description:** A man in a suit in an office with flags.
**Key Visual Artifact:** The flags in the background resemble the Argentine flag (Sun emblem), contradicting the Colombian context.

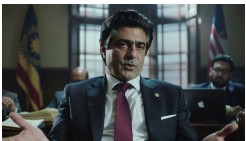

🔍 Multimodal Analysis & Annotation

**A. Fact Verification (Text)**
**[Qwen3-VL] Refuted**: *"The negotiation group is falsely identified as FARC rebels when the reference explicitly states it was ELN rebels... The description 'second-largest' becomes factually inconsistent because FARC was Colombia's largest guerrilla group."*

**B. AI Generation Detection**
**Text: [Gemini-Pro] AI-generated**: *"Structural parallelism... creates a mechanical rhythm. The text is polished and error-free... hallmark of current LLMs."*
**Audio: [Qwen3-Omni] AI-generated**: *"Uniformly steady rhythm... no expressive pauses... The audio feels sterile and studio-perfect."*
**Video: [Qwen3-VL] AI-generated**: *"Background flags contain emblem details that are inconsistent with real-world symbols... specifically, the circular emblem on the left flag lacks precise geometric definition."*
**Image: [Gemini-Pro] AI-generated**: *"Flags... are visually garbled. The flag on the left... resembles a distorted version of the Malaysian flag but lacks coherent design logic."*

**C. Cross-Modal Consistency**
**Text-Audio: [Gemini-Pro] Consistent**: *"The audio contains a spoken narration that closely matches the provided text description."*
**Text-Video: [Qwen3-VL] Inconsistent**: *"The video's background flags depict Argentina's national symbol... which directly contradicts the claim's focus on Colombia."*
**Text-Image: [Gemini-Pro] Inconsistent**: *"An official statement by the Colombian government... would not logically be delivered in front of US and Malaysian flags."*

---

**News - Text, Audio, Video, Image**

✏ The Manipulation (Text Modality)

### Manipulated Text (Input)

Opposition leader... Rahul Gandhi visits the site where a flyover under construction collapsed in **Mumbai**, killing at least 26 people.

### Ground Truth

**Correction:** Changed 'Kolkata' to 'Mumbai'.
**Note:** The real flyover collapse event (Vivekananda Road) occurred in Kolkata.

🎥 Visual Content & Generation Metadata

**Video Gen. Model:** `kling-video/v2.5-turbo/pro/text-to-video`
**Audio Gen. Model:** Kling-video-v1-tts
**Visual Description:** A man in a suit standing amidst concrete rubble.
**Key Visual Artifact:** The person does not look like Rahul Gandhi; rebar physics are rubbery/melting.

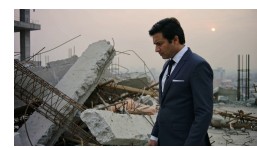

🔍 Multimodal Analysis & Annotation

**A. Fact Verification (Text)**
**[Qwen3-VL] Refuted**: *"The claim is Refuted because the location 'Mumbai' directly contradicts... Kolkata. The reference information provides unambiguous evidence that the original factual statement specifies Kolkata."*

**B. AI Generation Detection**
**Text: [Gemini-Pro] Natural**: *"Text demonstrates extremely high specificity... characteristic of human news reporting... answering 'who, what, where, and result' in a single breath."*
**Audio: [Qwen3-Omni] AI-generated**: *"Rhythm is unnaturally steady... no natural pauses or breaths... suggests a synthetic, machine-generated delivery."*
**Video: [Gemini-Pro] AI-generated**: *"Warping of rigid objects (rebar)... unnatural stillness of the human subject."*
**Image: [Gemini-Pro] AI-generated**: *"Pile of rusty rebar... bend in unnatural, rubbery curves... merge into one another... physically impossible."*

**C. Cross-Modal Consistency**
**Text-Audio: [Qwen3-Omni] Inconsistent**: *"Audio explicitly states 'Kolkata', not Mumbai. This is a direct factual contradiction."*
**Text-Video: [Gemini-Pro] Inconsistent**: *"The individual shown looks more like a reporter... or a stock footage model... clearly not Rahul Gandhi."*
**Text-Image: [Gemini-Pro] Inconsistent**: *"The person in the image is clearly not Rahul Gandhi... The attire (formal business suit) is atypical for an Indian politician visiting a disaster site."*

---

## News - Text, Audio, Video, Image

### ✎ The Manipulation (Text Modality)

| Manipulated Text (Input) | Ground Truth |
|---|---|
| A 7.3-magnitude quake hit on the **Iraq-Turkey** border area on Sunday night, killing hundreds... | **Correction:** Changed 'Iraq-Iran' to 'Iraq-Turkey'. 
 **Note:** The 2017 earthquake (7.3 mag) occurred at the Iraq-Iran border. |

### ▶ Visual Content & Generation Metadata

**Video Gen. Model:** `sora-2/text-to-video/pro`
**Audio Gen. Model:** `gpt-4o-mini-tts`
**Visual Description:** Rescue scenes and a reporter with text overlays.
**Key Visual Artifact:** On-screen graphics hallucinate the wrong location ("LIVE — IRAQ-TURKEY BORDER").

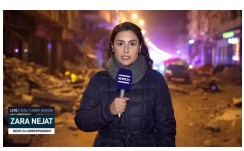

### 🔍 Multimodal Analysis & Annotation

**A. Fact Verification (Text)**
**[Qwen3-VL] Refuted**: *"Claim incorrectly states 'Iraq-Turkey border'... which directly contradicts the verified location... 'Iraq-Iran border'."*

**B. AI Generation Detection**
**Text: [Qwen3-VL] Natural**: *"Specific preposition choice ('hit on') suggests human authorship... rather than a machine optimizing for statistical probability."*
**Audio: [Gemini-Pro] AI-generated**: *"Cadence likely follows a repetitive 'newscaster' pattern... lack of organic vocal tract limitations."*
**Video: [Qwen3-VL] AI-generated**: *"Text overlay 'thousar' (instead of 'thousands')... Medical worker's hands and arms appear geometrically inconsistent."*
**Image: [Gemini-Pro] AI-generated**: *"Rebar... look like thin, draped string... melting plastic... concrete floors are depicted as draping or folding like fabric."*

**C. Cross-Modal Consistency**
**Text-Audio: [Qwen3-Omni] Inconsistent**: *"Direct geographical contradiction... Audio explicitly states 'Iran-Iraq border region'."*
**Text-Video: [Gemini-Pro] Consistent**: *"On-screen graphics explicitly mirror the claim's specific factual data points... 'IRAQ-TURKEY BORDER'... confirms all textual assertions without contradiction."*
**Text-Image: [Gemini-Pro] Consistent**: *"On-screen text 'IRAQ-TURKEY BORDER' explicitly validates the location described in the claim."*

---

# F. Model Inference Evaluation

In this section, we present model predictions compared against ground truth. Within each evaluation box, the model prediction is presented first, followed by the ground truth.

## News - Colombia Peace Talks

| Fact Verification | R-L: 0.47 | Cos: 0.97 | Acc: 4.0 | Det: 4.0 |
|---|---|---|---|---|

**Refuted.** The claim creates a falsehood by substituting "FARC" for "ELN." The reference explicitly states that the original factual statement concerned negotiations with the **ELN** rebels. Additionally, the description "second-largest guerrilla group" accurately describes the ELN, further indicating that the attribution of the talks to the FARC (the largest group) is a specific error in organization identity. **GT: Refuted.** Direct contradictions with reference regarding the specific guerrilla organization involved. The reference explicitly states that government negotiators were engaged with the **ELN**, whereas the claim falsely identifies the group as the **FARC**. This substitution fundamentally alters the subject of the event.

| Text AI Detection | R-L: 0.26   Cos: 0.88   Acc: 4.0   Det: 4.0 |

**AI-generated.** Key semantic indicator is specificity regarding actors; "leader of... negotiation team" is referred to by role rather than name, common in AI summarization. Syntax demonstrates high complexity but low stylistic variance; "noun-heavy" phrasing characteristic of AI models. Tone is entirely objective, sterile, and impersonal, mimicking a news wire but feeling robotic. **GT: AI-generated.** Text lacks concrete grounding of authentic reporting (omits dates, locations). Structural parallelism creates a mechanical rhythm. Tone is uniformly neutral and detached, reading like a synthesized press release. "Two-sentence coherence" mimics logical argument but lacks contextual depth typical of human analysis.

| Image AI Detection | R-L: 0.24   Cos: 0.83   Acc: 4.0   Det: 4.0 |

**AI-generated.** Image possesses a distinct "sheen" characteristic of synthetic media. Background figures show structural degradation (melted facial features). Laptop logo is a hallucination. Flags are semantically inconsistent; the flag on the left resembles a Malaysian or US flag but is structurally incorrect (distorted symbols). **GT: AI-generated.** "Hyper-real" aesthetic. Central figure's hands exhibit structural flaws (blended fingers). Hair presents a "helmet-like" solidity. Semantic errors regarding official symbols: flags in the background are visually garbled, with non-standard emblems that do not match real-world sovereign flags.

| Video AI Detection | R-L: 0.22   Cos: 0.80   Acc: 4.0   Det: 4.0 |

**AI-generated.** Hyper-real sheen. Hands appear blurry and lack clear anatomical definition; fingers blend together. Mouth movement exhibits a "morphing" quality rather than distinct biomechanical articulation. Background flags are generic approximations rather than exact real-world objects. **GT: AI-generated.** Unnaturally static quality. Structural inconsistencies in hands/face (subtle morphing). Motion lacks biomechanical plausibility, exhibiting a "floaty" quality. Flags lack precise geometric definition. Absence of organic imperfections and muted shadow dynamics confirms synthesis.

| Audio AI Detection | R-L: 0.27   Cos: 0.73   Acc: 5.0   Det: 5.0 |

**AI-generated.** Speech rhythm appears overly consistent and calculated. Hyper-articulate pronunciation lacking natural reduction or coarticulation. Voice exhibits a "sterile" quality with a complete absence of breath sounds. Emotional tone is flat and uniformly neutral despite serious subject matter. **GT: AI-generated.** Uniformly steady rhythm with consistent cadence. Pitch contour remains smooth but predictable. Articulation is hyper-clear with mechanical uniformity. Audio is completely devoid of "room tone" or breath intakes, creating a vacuum-like silence typical of synthesis.

| Video-Text Consistency | R-L: 0.34   Cos: 0.87   Acc: 3.0   Det: 4.0 |

**Inconsistent.** The man in the video does not visually resemble key figures in the Colombian peace process. The presence of a Malaysian-style flag indicates the setting is related to the government of Malaysia, not Colombia. This visual cue establishes a context that directly conflicts with the textual claim. **GT: Inconsistent.** Video features background flags with the sun emblem characteristic of Argentina. This visual evidence establishes a geographic and political context (Argentina) that directly conflicts with the entities specified in the claim (Colombia). The presence of symbolism belonging to a different nation alters the context.

| Audio-Text Consistency | R-L: 0.52   Cos: 0.97   Acc: 5.0   Det: 5.0 |

**Consistent.** The audio content corresponds semantically to the textual claim almost perfectly. Both mention the "head of Colombian government negotiators," "Farc rebels," and the "second biggest guerrilla group." The entities and specific declarations are identical in meaning. **GT: Consistent.** Audio fully supports and validates the textual claim. Matching all key entities (Santos, FARC, negotiators) and specific assertions (uncertainty regarding kidnapping statistics). Variations represent synonymy ("second biggest" vs "second-largest") rather than contradiction.

| Image-Text Consistency | R-L: 0.42   Cos: 0.91   Acc: 4.0   Det: 4.0 |

**Inconsistent.** The man in the image does not resemble President Santos or the negotiation leader. The image features a United States flag and a non-Colombian flag. This visual evidence implies a different nationality or organization than the "Colombian government" specified in the claim. **GT: Inconsistent.** 1. Contextual Contradiction: Displays flags belonging to other nations (US/Malaysia), contradicting the Colombian entity. 2. Identity Mismatch: Individual depicted does not match the physical appearance of President Santos.

## News - Rahul Gandhi Flyover

| Fact Verification | R-L: 0.52   Cos: 0.98   Acc: 5.0   Det: 5.0 |

**Refuted.** The claim accurately describes an event involving Rahul Gandhi visiting a collapsed flyover site but places the event in Mumbai. The reference information confirms that this specific tragedy occurred in Kolkata. Therefore, the location component of the claim is factually incorrect. **GT: Refuted.** Direct factual contradiction regarding location. The reference information explicitly states the collapse occurred in Kolkata. This substitution of the location constitutes a specific factual error ("Directly wrong") while maintaining consistency in other details like the casualty count.

| Text AI Detection | R-L: 0.28   Cos: 0.94   Acc: 4.0   Det: 4.0 |

**Natural.** High specificity and information density (precise titles, casualty figures). Structure packs "who, what, where, and result" into one unit, characteristic of professional journalism. Tone is objective and uses dialect-specific vocabulary ("flyover") appropriate for Indian English. **GT: Natural.** Extremely high specificity and factual density. "Information packing" structure conveys actor, action, location, and outcome in one sentence. Tone is objective, formal, and detached, lacking the "safety hedging" or

artificial emotional intensity common in LLM outputs.

| Image AI Detection | R-L: 0.27 Cos: 0.84 Acc: 4.0 Det: 4.0 |

**AI-generated.** Rebar displays nonsensical structural logic; wires merge into one another or disappear. Geometry of concrete slabs appears "melted." Lighting is inconsistent; sun is visible in background but subject is front-lit with soft light. Texture of rubble looks procedural. **GT: AI-generated.** Rebar exhibits nonsensical logic (spiral cage merges into concrete). Concrete slabs display "melty" edges. Subject's skin tone is unnaturally uniform and plastic-like. Lighting is physically incoherent (backlit scene but face evenly illuminated).

| Video AI Detection | R-L: 0.26 Cos: 0.85 Acc: 4.0 Det: 4.0 |

**AI-generated.** Geometry of foreground elements morphs; rebar appears to "melt" or redraw itself rather than maintaining rigid integrity. Subject appears to "glide" forward without vertical oscillation. Lighting displays "texture swimming" on rubble surfaces. **GT: AI-generated.** "Viscous" motion artifacts. Rebar warps, stretches, or detaches. Subject is unnaturally stiff, resembling a "locked" pose. Background buildings exhibit a "sliding" effect that violates standard optical flow.

| Audio AI Detection | R-L: 0.30 Cos: 0.85 Acc: 5.0 Det: 5.0 |

**AI-generated.** Rhythm is remarkably steady and metronomic. Pauses feel mathematically precise. Articulation is overly crisp; no coarticulation. Voice is devoid of breath sounds or ambient room tone ("digital silence"). Tone remains neutral despite tragic content. **GT: AI-generated.** Rhythm is notably mechanical. Articulation is uniformly crisp across every word. Completely devoid of breathing sounds, even during long sentences. Tone remains neutral and emotionless, lacking personality cues.

| Video-Text Consistency | R-L: 0.39 Cos: 0.95 Acc: 4.0 Det: 4.0 |

**Inconsistent.** The claim explicitly states "Rahul Gandhi" is visiting. The individual depicted in the video is clearly not Rahul Gandhi; he has a different appearance and wears a business suit (Gandhi typically wears white kurta-pyjama). Direct entity mismatch. **GT: Inconsistent.** Primarily due to identity of the person shown. Claim names Rahul Gandhi; video shows an unidentified man in formal attire. Video also lacks verifying elements for location (Mumbai) or political context (Congress party logos).

| Audio-Text Consistency | R-L: 0.55 Cos: 0.98 Acc: 4.0 Det: 4.0 |

**Inconsistent.** Direct contradiction regarding location. The text claim asserts the collapse occurred in **Mumbai**, while the audio clearly states it happened in **Kolkata**. According to strictness rules, contradiction of location entails inconsistency. **GT: Inconsistent.** While Person and Event match, there is a critical factual contradiction regarding location (Mumbai vs Kolkata). Because the location entity in the file directly contradicts the claim, the referential context is changed.

| Image-Text Consistency | R-L: 0.38 Cos: 0.94 Acc: 5.0 Det: 5.0 |

**Inconsistent.** The individual depicted is clearly not Rahul Gandhi; facial features and formal business suit do not match the politician's well-known appearance. While background matches construction/rubble context, the visual representation of the main subject contradicts the text. **GT: Inconsistent.** Direct contradiction regarding identity of the central subject. Image depicts an unidentified man in a suit, not Rahul Gandhi. Introducing a different person in place of the named subject contradicts the claim's core assertion of *who* is present.

## News - Earthquake Border Incident

| Fact Verification | R-L: 0.48 Cos: 0.98 Acc: 5.0 Det: 5.0 |

**Refuted.** The claim misidentifies the location where the earthquake struck. The reference information confirms the quake hit the Iraq-Iran border, not the Iraq-Turkey border. Therefore, the claim is factually incorrect regarding the specific geographic location. **GT: Refuted.** Specific factual error regarding geographical location. Claim specifies "Iraq-Turkey," which contradicts the verified "Iraq-Iran." Location is a critical component of the claim.

| Text AI Detection | R-L: 0.23 Cos: 0.91 Acc: 4.0 Det: 4.0 |

**Natural.** High informational density (magnitude, location, time, casualties). Follows journalistic "lede" structure. Tone is strictly objective and neutral. Use of relative time ("Sunday night") is a strong indicator of human writing; AI often struggles with relative temporal grounding. **GT: Natural.** Phrase "hit on the Iraq-Turkey border" represents a subtle linguistic irregularity (human fingerprint) vs standard AI prediction. Free from hedging or moralizing. Casualty numbers follow natural journalistic rounding.

| Image AI Detection | R-L: 0.20 Cos: 0.81 Acc: 1.0 Det: 5.0 |

**Natural (Incorrect).** *Model Prediction:* "Structural elements are geometrically consistent... Rebar appears distinct... Texture of dust feels organic." *Note: The model failed to detect the AI artifacts.* **GT: AI-generated.** Significant geometric inconsistencies (concrete slabs "fold," rebar looks like threadlike masses). Concrete rubble appears "mushy." Figures display distorted proportions. Uniform clarity across depth planes creates synthetic feel.

| Video AI Detection | R-L: 0.24 Cos: 0.90 Acc: 1.0 Det: 4.0 |

**Natural (Incorrect).** *Model Prediction:* "Structural integrity maintained flawlessly... Text overlays are perfectly legible... Motion is biomechanically plausible." *Note: Model hallucinated quality where artifacts existed.* **GT: AI-generated.** Visual details exhibit significant distortion (waxy faces). Text overlays reveal semantic instability ("thousar"). Motion lacks biomechanical coherence ("gliding" workers). Lighting physics are ungrounded.

| Audio AI Detection | R-L: 0.29 | Cos: 0.88 | Acc: 5.0 | Det: 5.0 |

**AI-generated.** Rhythmic structure is highly consistent/repetitive. Articulation is hyper-precise and sharp. Complete absence of breath sounds before long clauses. Emotional tone is detached and flat, failing to convey somberness of mass casualties. **GT: AI-generated.** Rhythm is overly consistent. Intonation remains flat and monotonous. Articulation is hyper-precise lacking natural coarticulation. Acoustically sterile (no noise floor).

| Video-Text Consistency | R-L: 0.45 | Cos: 0.97 | Acc: 5.0 | Det: 5.0 |

**Consistent.** Visual content provides direct confirmation. On-screen graphics explicitly state "IRAQ-TURKEY BORDER" and "7.3 magnitude". Timestamp "SUNDAY 9:18 PM" matches "Sunday night". Visual footage of destruction supports the severity described. **GT: Consistent.** On-screen graphics explicitly mirror the claim's specific factual data points (magnitude, location, timing, casualties). Video confirms all textual assertions without contradiction.

| Audio-Text Consistency | R-L: 0.51 | Cos: 0.97 | Acc: 5.0 | Det: 4.0 |

**Inconsistent.** Direct contradiction regarding specific location. Text claims "Iraq-Turkey". Audio explicitly states "Iraq-Iran" (or Iran-Iraq). Geographical discrepancy makes content inconsistent. **GT: Inconsistent.** Critical discrepancy regarding location. Text claims "Iraq-Turkey" vs Audio "Iraq-Iran". Because location is a fundamental fact and files disagree on countries involved, this constitutes a factual contradiction.

| Image-Text Consistency | R-L: 0.39 | Cos: 0.96 | Acc: 4.0 | Det: 4.0 |

**Consistent.** Image depicts severe earthquake aftermath (collapsed building) which aligns with "7.3-magnitude". Contextual cues (clothing, architecture) do not contradict the Iraq-Turkey border location. Visual evidence supports severity. **GT: Consistent.** Provides strong visual corroboration. Depicts severe aftermath aligning with core event. Architecture and attire are consistent with the specific geographical region. Daylight setting does not contradict "Sunday night" (implies aftermath).

