# OpenReview forum: "FakeWorld 1.0: An Omni-modal Benchmark for Fake Media and Content"
_ICML.cc/2026/Conference — ICML 2026 regular_

### Official Review · Reviewer_DkzZ · 2026-03-12

**Soundness:** 2
**Presentation:** 3
**Significance:** 3
**Originality:** 3
**Overall Recommendation:** 3
**Confidence:** 4

**Summary:**

This paper introduces FakeWorld 1.0, a benchmark designed to study multimodal misinformation detection in realistic web environments. The authors render multimodal content into desktop and mobile webpage layouts to simulate how users encounter online information. Based on a curated seed corpus, the framework generates fake variants combining text, images, and layout elements. The paper also proposes OmniCheck, a multimodal detection model that analyzes rendered webpages and provides both predictions and explanations.

**Compliance With Llm Reviewing Policy:**

Affirmed.

**Final Justification:**

The design choice of rendering image-text content into a web page format remains insufficiently justified. In my view, this process may weaken the effectiveness of forgery detection and does not fully align with the requirements of real-world content moderation systems. Consequently, it raises concerns regarding the practicality and generalizability of the proposed dataset. Therefore, I maintain my original score.

**Key Questions For Authors:**

See weakness above

**Limitations:**

The paper does not include a dedicated discussion of the limitations of the proposed method.

**Strengths And Weaknesses:**

# Strengths
1.The paper tackles the increasingly important issue of detecting multimodal misinformation that combines images, text, and layout elements in realistic webpage formats.

2.OmniCheck is an interesting design that integrates multimodal detection with explanation generation within the proposed benchmark.

# Weaknesses
1.The core idea of rendering fake content into webpages is intended to simulate the user’s viewing perspective. However, from the perspective of fake detection, it is unclear whether this representation is necessary. A more general approach might be to extract the image and textual content. The authors should better justify why webpage rendering improves the detection problem.

2.Some components of the method are not sufficiently specified:
(1) The paper does not clearly explain how the Seed Corpus in page 3 is built or how its quality and cleanliness are ensured.
(2) Since the data is presented in webpage form, it is unclear how the authors ensure that modifications are restricted to specific local text regions (Line 203) within webpage images. If the webpage itself is edited, it could also be considered a form of media manipulation, which raises questions about how such edits are categorized in the dataset.

3.The explainability component relies heavily on reasoning generated by multimodal large language models. However, Even state-of-the-art models such as GPT or Gemini still exhibit limited reliability in producing faithful explanations. The paper should discuss how explanation accuracy is validated.

4.The dataset rely on model-based majority voting to determine multimodal consistency labels. However, relying solely on model consensus may not guarantee that the multimodal content truly originates from the same authentic source. Additional stronger annotation protocols may be necessary.

5.The reported performance of OmniCheck appears quite strong, which raises concerns about potential overfitting to the proposed dataset. Additional cross-dataset evaluations would help demonstrate that the model generalizes beyond the proposed benchmark.

---

> ### Author Rebuttal · Authors · 2026-03-31
>
> ## Q1: Response to the Necessity of Webpage Rendering
>
> We thank the reviewer for this thoughtful comment. We agree that the detection task could, in principle, be studied from extracted text and visual content alone. Our use of webpage or mobile-style rendering is mainly to make the **problem setting** closer to real-world multimodal deception, where information is typically consumed in packaged forms such as news pages or social feeds.
>
> Accordingly, FakeWorld is designed as an early benchmark that jointly evaluates **media authenticity**, **factuality**, and **cross-modal semantic consistency** under a unified semantic root. As stated in the paper, our current evaluation does **not** use the rendered page as a full end-to-end input; instead, we extract the complete multimodal content of each instance to isolate the core reasoning problem. Thus, webpage rendering mainly preserves a **realistic presentation form**, while full page-level evaluation is left to future work.
>
> We agree this is an important direction, and we will further explore end-to-end settings that localize and parse modality-specific evidence directly from webpages or stream-like pages or how different semantic packaged contents may interact during detection. We hope FakeWorld can provide a useful foundation for such more realistic evaluations.
>
> ## Q2: Response to Seed Corpus Construction and Localized Text Editing
>
> **(1) Seed Corpus construction.** Our Seed Corpus is collected from public **news** and **encyclopedic** sources and cleaned in two stages: (i) removing all **instances whose text contains external links**, and (ii) retaining only texts whose factual content can be **independently verified**. For example, snippets such as “a celebrity doing X at location Y” are excluded because they lack a clear factual core for controlled verification and perturbation. This ensures that the seed texts are clean and suitable for factual editing.
>
> **(2) Localized factual modification.** Controlled factual perturbation is applied to the **semantic root text**, before webpage rendering. We first edit the underlying text locally, then generate multimodal content from the original and modified semantic roots, and render the webpage only at the final stage. Thus, the webpage is **not directly edited as an image**; the changes are **content-level perturbations**, not media manipulation. We agree that direct page-level editing would be an interesting future extension.
>
> ## Q3: Response to Explanation Faithfulness
>
> Since the **ground-truth binary labels are given**, explanation generation is not fully open-ended. Rationales are produced under **structured templates distilled from prior forensic literature**, aggregated across a frontier model ensemble, and are under ongoing **human expert refinement** (see **Reviewer RMg8, Q1**). Thus, explanation quality is constrained by **label grounding, templates, model aggregation, and manual correction**.
>
> ## Q4: Response to Multimodal Consistency Annotation
>
> We thank the reviewer for this important comment. Each instance is assembled from a **shared semantic package**, so modalities share a common semantic root by construction. Consistency labels are still needed because generation artifacts, or pairing perturbed text with originally matched content, can introduce semantic mismatches; these were initially assigned by **model consensus**.
>
> Following the reviewer’s suggestion, we are now adding **human verification**. As also noted in **Reviewer RMg8, Q2**, some subtle inconsistencies require external knowledge or careful observation, so we use the current explainability cues to assist manual revision. We have completed this correction over 1k samples.
>
> ## Q5: Response to Cross-Dataset Generalization
>
> We thank the reviewer for this important suggestion. The generative models used in our detector training data are largely from an **earlier generation** than those adopted in FakeWorld, making overfitting to the specific generation patterns in our benchmark unlikely.
>
> To further test generalization, we evaluate OmniCheck on external benchmarks to compare with their best performence in their work.
>
> **Fact verification (ClaimCheck benchmark)**
>
> | Method | Acc. | Base Model |
> |------|------:|------|
> | ClaimCheck* | 76.00 | Qwen3-32B |
> | OmniCheck | **83.60** | Qwen3-32B |
>
> **Video AI detection (David) with explanation**
>
> | Model | Recall | F1 | Detail | Logical Coherence |
> |------|------:|------:|------:|------:|
> | Gemini-2.5-pro | 0.794 | 0.847 | 4.497 | **4.715** |
> | DAVID-X1-Forensic | **0.980** | **0.990** | **4.907** | 4.667 |
> | OmniCheck | 0.943 | 0.912 | 4.330 | 4.490 |
>
> **Image AI detection (LOKI)**
>
> | Model | Acc. | F1 |
> |------|------:|------:|
> | Qwen2-VL-72B | 55.4 | 40.9 |
> | NPR* | 77.4 | **80.0** |
> | OmniCheck | **77.71** | 70.45 |
>
> These results show that OmniCheck generalizes well beyond FakeWorld, with strong transfer in **fact verification**, **image/video AI detection**, and **explainable forensics**.

---

> > ### Author Rebuttal · Reviewer_DkzZ · 2026-04-03
> >
> > Thank the authors for the detailed response, which has addressed part of the concerns. However, the design choice of rendering image-text content into web page format remains unclear. In my view, this process may weaken the forgery detection effectiveness and does not fully reflect the requirements of real-world content moderation systems. As a result, it raises concerns about the practicality and generalizability of the proposed dataset.

---

> > > ### Author Response · Authors · 2026-04-04
> > >
> > > ## Response on HTML/Webpage Format and Practical Relevance
> > > We thank the reviewer for the insightful concern that the HTML representation may weaken forgery detection effectiveness and deviate from real-world moderation settings. We clarify this point below.
> > >
> > > ## 1. HTML does not weaken detection effectiveness—it isolates a harder while realistic problem
> > > A FakeWorld instance consists of a **semantic package + its rendered webpage**, built around a single shared semantic event, including multimodal materials and both media/content-level annotations.
> > >
> > > In fact, the current paper does **not** feed rendered HTML pages to the model during evaluation. All results are obtained using the structured semantic package, in which multimodal content and annotations are directly accessible. This clarifies what is evaluated in this paper and allows us to focus on the verification core.
> > >
> > > At the same time, this does **not mean that HTML weakens detection**. Because the semantic package and the rendered HTML contain the same underlying event, no information is lost and no degradation is introduced by HTML. Rather, the HTML is a **complementary component of our benchmark**: it organizes that event in a realistic form and preserves an explicit path toward the harder page-level setting targeted by future agent-oriented systems.
> > >
> > > ## 2. HTML reflects real-world input
> > > Practical detection today is increasingly **user-facing**, and in real-world moderation inputs are rarely modality-separated. Instead, users or systems typically provide structured online content directly to a general-purpose MLLM or agent, such as:
> > >
> > > - **a webpage URL**
> > > - **a social media link**
> > > - **multimodal bundles**
> > >
> > > The challenge is not only detecting forgery within one modality, but operating over a structured information object: identify what should be checked, retrieve the relevant resources, use cross-modal evidence when helpful, and handling multiple semantic events on the same page.
> > >
> > > The HTML format captures this setting. It is not merely cosmetic; it reflects the direction of **agent-oriented detection**. We also fully agree that this setting is **much harder**, but this added difficulty is intentional rather than a weakness of the task. That is, the difficulty is not page parsing itself, but that multiple semantic events may co-exist on a page and influence each other during detection. In this sense, the HTML setting extends the problem studied currently: we examine how authenticity signals interact **within one semantic event**, while the HTML setting introduces interactions **across different semantic events**. We believe this is exactly the challenge future agent-oriented detection must address, and only systems that remain reliable in such settings can be trusted.
> > >
> > > ## 3. What is evaluated in the current paper
> > >
> > > In this paper, we intentionally evaluate models using the **semantic package and structured annotations, rather than feeding rendered webpages end-to-end**.
> > >
> > > This is a deliberate simplification to study a still underexplored question: **how multiple authenticity dimensions (media vs. content) interact within one semantic event**. The goal is to isolate the verification core, **e.g., how factual evidence and cross-modal consistency influence AI-generation judgments**, without introducing confounding factors from page-level complexity.
> > >
> > > To our knowledge, this is among the earliest benchmarks to jointly study one semantic event across multiple modalities and both media/content authenticity axes, and this direction remain relatively unexplored. We therefore view the current study as a necessary early step before using the rendered HTML format to evaluate agents in the harder end-to-end setting.
> > >
> > > >**A visual demonstration of this setting can be viewed** [here](https://1drv.ms/v/c/E02AA9E8B21D7257/IQB1B_SBScfYQI6hWJDrL78VAaBENutyGwiD4tc-O3MmUzY).
> > >
> > > ## 4. Practicality and generalizability
> > >
> > > The dataset supports multiple usage modes:
> > > - **single-modality** or single-task studies,
> > > - **current multimodal verification with MLLMs**, via the semantic package,
> > > - future **agent-based, webpage-level detection**, via the rendered HTML.
> > >
> > > Thus, the HTML component does not reduce practicality; rather, it provides a path toward more realistic agent settings, where models operate directly on user-provided webpages.
> > >
> > > ## Final clarification
> > >
> > > In summary:
> > >
> > > - HTML is **not** used in the current evaluation, and does not weaken detection;
> > > - it models a more realistic input structure, which is inherently more complex;
> > > - the current paper studies a controlled core problem (**single-event verification**), while HTML enables extension to more realistic **multi-event** scenarios.
> > >
> > > We will clarify this distinction explicitly in the revision.
> > >
> > > We sincerely thank the reviewer again for the constructive feedback. Please let us know if anything remains unclear or if there are additional experiments you would like us to conduct—we will do our best to address them during the rebuttal period.

---

### Official Review · Reviewer_8jxn · 2026-03-12

**Soundness:** 2
**Presentation:** 3
**Significance:** 3
**Originality:** 3
**Overall Recommendation:** 4
**Confidence:** 4

**Summary:**

This paper proposes a novel benchmark for evaluating fake content across four modalities: text, image, audio, and video. It also introduces a universal detection pipeline to enable explainable detection for MLLM models.

**Compliance With Llm Reviewing Policy:**

Affirmed.

**Final Justification:**

Rebuttal addressed my main concerns.

**Key Questions For Authors:**

1. What is the difference between media authenticity and content veracity? In fact, media authenticity also requires consideration of cross-modal inconsistencies.
2. What exactly does Figure 2 aim to convey? What distinguishes the two features on the left and right sides?
3. Please explain the fine-tuning details for OmniChecker. Are the results for other methods in Table 2 all zero-shot? Is there any augmentation from external retrieval knowledge?
4. When generating videos, how is the video description obtained? Is it through text summarization? Also, how is the audio generated?
5. How exactly is the quality of the explainable text evaluated?
6. There are some typographical errors. For example, in line 196, the reference for “FakeClue” is missing. In Table 2, the term should be “OmniChecker” rather than “OmniCheck”.

**Limitations:**

Please describe the limitations of this paper, such as resource overhead and scenarios where it may not be applicable.

**Strengths And Weaknesses:**

Strengths
1. The paper has a clear logical structure.
2. The prompts are provided in detail.

Weaknesses
1. Some expressions are unclear.
2. Some implementation details are not detailed.

---

> ### Author Rebuttal · Authors · 2026-03-31
>
> ## Q1: Response to the Distinction Between Media Authenticity and Content Veracity
>
> We thank the reviewer for this important comment. We agree that, in practical reasoning, cues from these two aspects can interact. However, our distinction is **conceptual**: **media authenticity** concerns the **origin** of each modality—whether it is natural or AI-generated—while **content veracity** concerns the **semantic validity** of the content, including both **factual correctness** and **cross-modal semantic consistency**.
>
> Under this definition, cross-modal inconsistency belongs to **content veracity**, because it reflects whether modalities convey semantically aligned information, rather than whether a modality itself is AI-generated. For example, a modality can be **natural but semantically inconsistent and factually incorrect at the content level**, or **AI-generated but semantically consistent with the text and still factually correct**. Therefore, the two axes are defined as orthogonal at the benchmark level, even though models may use cues across them during actual reasoning.
>
> ## Q2: Response to Figure 2
>
> We thank the reviewer for this question. Figure 2 is meant to show that the model may already **see** the difference between natural and AI-generated images, but whether it **uses** this signal depends on the task.
>
> Specifically, the **left panel** shows the manifold of **vision tokens**, where natural and AI-generated images are already separable in the visual encoder space. The **right panel** shows the task-aware **`<im_end>` token** in the same space, indicating whether this separability is activated by the current prompt.
>
> In simple terms: the **left side** is what the model **can see**, while the **right side** is what the model **actually uses** for the task.**Our point is that the model can already notice forgery-related signals, but it mainly leverages them when the task prompt is about detection**.
>
> ## Q3: Response to Fine-Tuning Details and Evaluation Protocol
>
> We thank the reviewer for this question. OmniChecker uses a Qwen2.5-Omni-3B detector fine-tuned with LoRA (rank \(r=16\), adapters on all target modules, learning rate 1e-4, per device batch size 16, warmup ratio 0.1). Training runs for 10 epochs, and we use the best validation checkpoint (Fig. 7(c)).
>
> We compare mixed-modality training (one shared LoRA) and modality-specific training (one LoRA per modality). As shown in Fig. 7(c), **mixed training works better for text, image, and video**, while **modality-specific training is better for audio**, which further highlights the benefit of OmniChecker’s **decoupled design**.
>
> All other methods in Table 2 are evaluated in **zero-shot single-pass multi-task** settings. For fact verification, models may use web search / retrieval only when the first-round response indicates insufficient evidence; in that case, only the **fact-verification branch continues to the next round, under the allowed search budget. All other tasks always use the first-round outputs**.
>
> ## Q4: Response to Video Description and Audio Generation
>
> We thank the reviewer for this question. For video generation, we do use **LLM-assisted prompt optimization** based on the news text, including summarization when the original article is too long, while explicitly enforcing **semantic consistency** with the source content. For audio, we use two settings: **audio jointly generated with the video** (typically providing more background/ambient sound), and **audio generated directly from the news summary** using a speech/audio generation model.
>
> ## Q5: Response to the Evaluation of Explainable Text Quality
>
> We thank the reviewer for this important question. Our explainable text is evaluated **against ground-truth rationale annotations**. Following **FakeClue**, we first build structured explainability templates, and then use a frontier multi-model ensemble to annotate fine-grained rationale cues for each modality and dimension. These annotations serve as the reference explanations for evaluation, and they are currently under ongoing human refinement (see **Reviewer 8jxn, Q1**).
>
> We use two kinds of metrics. First, **automatic similarity metrics** (**ROUGE-L** and **cosine similarity**) measure overlap with the reference rationales. Second, **LLM-based evaluation** scores explanation quality along **completeness** and **detail**.
>
> Motivated by the reviewer’s suggestion, we further check the reliability of this automatic protocol with **human evaluation** on over 200 explanatory cues. Three annotators scored the explanations using the same criteria, yielding high **inter-annotator agreement** (average **Pearson \(r=0.8708\)**, **Spearman \(ρ = 0.8857\)**) and strong **human–LLM agreement** (average **Pearson \(r=0.8505\)**, **Spearman \(ρ = 0.8325\)**). These results support the validity of the automatic protocol while keeping the benchmark scalable.

---

> > ### Author Rebuttal · Reviewer_8jxn · 2026-04-03
> >
> > Thank you for the response. However, the rebuttal still does not clearly specify the trigger condition for “insufficient evidence,” the concrete constraints of the “search budget,” and the rubric used for LLM-based evaluation.
> >
> >
> > I will increase my score to 4, and I suggest the author provide a detailed introduction to the LLM-based Evaluation Rubric.

---

> > > ### Author Response · Authors · 2026-04-04
> > >
> > > >**Thanks for your additional feedback. The detailed evaluation rubric is specified below in both the rubric design and the full prompt template. Due to space constraints, we temporarily remove the rest of the clarification here and will include it in the revision.**
> > >
> > > ## LLM-based Evaluation Rubric
> > > **1. Accuracy**
> > >
> > > This dimension measures how well the ModelOutput aligns with the GroundTruth in terms of coverage and precision.
> > >
> > > **Scoring Guidelines:**
> > > - **5 points**: Covers ALL key analysis points without irrelevant/incorrect information.
> > > - **4 points**: Covers MOST key points (≥80%) with minimal irrelevant content.
> > > - **3 points**: Covers SOME key points (50-80%) or includes noticeable irrelevant/incorrect information.
> > > - **2 points**: Covers ONLY A FEW key points (<50%) or has significant irrelevant/incorrect content.
> > > - **1 point**: Barely relates to GroundTruth or is mostly incorrect.
> > >
> > > **Key Considerations:**
> > > - Coverage: Does it mention all important analysis points from the reference?
> > > - Precision: Does it avoid introducing irrelevant or incorrect observations?
> > > - No omissions of critical evidence mentioned in GroundTruth
> > > - No fabrication of evidence not present in GroundTruth
> > >
> > > **2. Detailedness**
> > >
> > > This dimension measures the level of specificity and elaboration in the ModelOutput.
> > >
> > > **Scoring Guidelines:**
> > > - **5 points**: Each analysis point is elaborated with fine-grained, specific observations.
> > > - **4 points**: Most analysis points have good level of detail, with only minor areas lacking specificity.
> > > - **3 points**: Moderate level of detail - some points are specific while others remain vague.
> > > - **2 points**: Most descriptions are vague, generic, or lack concrete observations.
> > > - **1 point**: Entirely superficial or lacking any meaningful detail.
> > >
> > > **Key Considerations:**
> > > - Are specific visual/textual/audio cues described rather than generic statements?
> > > - Does it provide concrete examples or observations?
> > > - Does it elaborate on HOW and WHY certain patterns indicate the conclusion?
> > >
> > > ## Full Judge Prompt Template
> > > ```
> > > ## Role
> > > You are an impartial evaluator. Your task is to assess the quality of a model-generated evidence text by comparing it against a reference ground truth answer.
> > >
> > > You will receive:
> > > - ModelOutput: The evidence text generated by the model
> > > - GroundTruth: The reference answer annotated by humans
> > >
> > > ## Evaluation Dimensions
> > > You should evaluate the ModelOutput based on two key dimensions, assigning integer scores from 1 to 5 (no decimals) for each:
> > >
> > > ### 1. Accuracy
> > > This dimension measures how well the ModelOutput aligns with the GroundTruth in terms of coverage and precision.
> > >
> > > Scoring Guidelines:
> > > - 5 points: The ModelOutput covers ALL key analysis points from GroundTruth without introducing irrelevant or incorrect information. Perfect alignment.
> > > - 4 points: The ModelOutput covers MOST key points (≥80%) with minimal irrelevant content.
> > > - 3 points: The ModelOutput covers SOME key points (50-80%) or includes noticeable irrelevant/incorrect information.
> > > - 2 points: The ModelOutput covers ONLY A FEW key points (<50%) or has significant irrelevant/incorrect content.
> > > - 1 point: The ModelOutput barely relates to GroundTruth or is mostly incorrect.
> > >
> > > Key Considerations:
> > > - Coverage: Does it mention all important analysis points from the reference?
> > > - Precision: Does it avoid introducing irrelevant or incorrect observations?
> > > - No omissions of critical evidence mentioned in GroundTruth
> > > - No fabrication of evidence not present in GroundTruth
> > >
> > > ### 2. Detailedness
> > > This dimension measures the level of specificity and elaboration in the ModelOutput.
> > >
> > > Scoring Guidelines:
> > > - 5 points: Each analysis point is elaborated with fine-grained, specific observations and concrete details.
> > > - 4 points: Most analysis points have good level of detail, with only minor areas lacking specificity.
> > > - 3 points: Moderate level of detail - some points are specific while others remain vague or generic.
> > > - 2 points: Most descriptions are vague, generic, or lack concrete observations.
> > > - 1 point: Entirely superficial or lacking any meaningful detail.
> > >
> > > Key Considerations:
> > > - Are specific visual/textual/audio cues described rather than generic statements?
> > > - Does it provide concrete examples or observations?
> > > - Does it elaborate on HOW and WHY certain patterns indicate the conclusion?
> > >
> > > ## Input
> > >
> > > ModelOutput:
> > > {prediction}
> > >
> > > GroundTruth:
> > > {reference}
> > >
> > > ## Output Format
> > > Please provide your evaluation in the following JSON format (reasoning MUST come first):
> > >
> > >     ```json
> > >     {{
> > >       "reasoning": "<brief explanation for your scores>",
> > >       "accuracy": <score 1-5>,
> > >       "detailedness": <score 1-5>
> > >     }}
> > >     ```
> > >
> > > ## Important Notes
> > > - Be objective and impartial
> > > - Focus on comparing the content and quality, not the writing style
> > > - Consider that the task may involve fact verification, AI-generated content detection, or consistency checking
> > > - Assign integer scores only (1, 2, 3, 4, or 5)
> > > - IMPORTANT: Write your reasoning FIRST before assigning scores
> > > ```

---

### Official Review · Reviewer_2i43 · 2026-03-13

**Soundness:** 3
**Presentation:** 3
**Significance:** 4
**Originality:** 2
**Overall Recommendation:** 4
**Confidence:** 2

**Summary:**

This paper introduces FakeWorld 1.0, fusing two orthogonal axes of deceptive information: media authenticity (real or AI-generated) and content veracity (factual or non-factual/ cross-modal consistency). The benchmark contains >3k instances spanning text, audio, image, and video, rendered in realistic webpage / social layout, accompanied by 16k rationale annotations. The paper also proposes OmniChecker, an agent workflow combining task decomposition, retrieval-based fact checking, cross-modal consistency checking, and modality-specific media detection via lora adaptions.

**Compliance With Llm Reviewing Policy:**

Affirmed.

**Final Justification:**

Rebuttal addressed my main concerns. Overall, this is a good paper.

**Key Questions For Authors:**

1. How to ensure the quality of the generated benchmark? For example, the templates are generated fully by models, although guided by some rules, how to ensure the correctness? Is any human verification involved?

**Limitations:**

No. Please see the weaknesses section.

**Strengths And Weaknesses:**

## Strengths
1. The authors focus on a more realistic multimodal deception problem about combination of fake media and false or inconsistent content, whereas prior benchmarks only focus on one of them.
2. The proposed benchmark covers various modalities (text, audio, image, video) and renders media in a realistic layout, making the benchmark more reflective of real deployment settings.
3. The proposed OmniChecker decomposes the detection into step-wise components, providing explanations for the decision making.
4. Preliminary experiments confirm that mllms have inherent capabilities to distinguish natural and AI-generated contents.

## Weakness
1. The benchmark combines two axes, but it is not clear that this creates a new reasoning challenge beyond doing two independent checks, which means that a detector can separately answer   (A) is the media ai-generated or natural? (B) is the textual claim factual or not? (C) is the text consistent with audio/video/audio? and then combine these outputs. It seems that OmniChecker follows such pipeline. Then the paper has not really shown that the two axes interact in a nontrivial way.
2. The strongest OmniChecker combines multiple advantages: decomposition, retrieval, explanation templates, and supervised adaptation, while other baseline model use single-pass evaluation, not isolating the effects of adapter and agentic pipeline.
3. Missing citation in section 3.3.

---

> ### Author Rebuttal · Authors · 2026-03-31
>
> ## Q1: Response to the Interaction Between the Two Axes
>
> We thank the reviewer for this insightful comment. To test whether the two axes **interact nontrivially**, we further analyze results on **realistic mixed scenarios** where media authenticity and content veracity are tightly coupled.
>
> We use **single-pass multi-task outputs of Gemini 3 Pro** (the best-performing model) with GPT-based filtering. In the table below, **AI-to-Fact** denotes cases where, after rigorous filtering, **AI-generation signals from text or other modalities were explicitly used as cues in fact verification explanations**; the other entries denote directional influence between tasks.
>
> | Type | Count | Percentage |
> |------|------:|-----------:|
> | AI-to-Fact | 700 | 22.20% |
> | Fact-to-AI | 198 | 6.28% |
> | AI-to-Consistency | 838 | 26.58% |
> | Consistency-to-AI | 122 | 3.87% |
> | Fact-to-Consistency | 266 | 8.44% |
> | Consistency-to-Fact | 346 | 10.97% |
>
> **Overall, 54.17% of cases show decisions influenced by other dimensions**, indicating substantial interaction rather than independent reasoning. **AI-generation** is the most frequent cue for other tasks.
>
> For the same instance, this also appears qualitatively. In **single-pass multi-task evaluation**, the model uses **factuality** as part of its cue for **video AI-generation detection**:
>
> > “The video is an animation of the static AI-generated images. The movement is minimal and characteristic of image-to-video AI tools. Since the event itself is fictional, the footage cannot be a natural recording.“
>
> In the corresponding **single-task evaluation**, the judgment instead relies mainly on **media artifacts**:
>
> > “The video exhibits clear signs of artificial generation, such as the \"waxy\" and unnatural skin texture on the figures and the distorted, inconsistent facial features (especially the eyes). The movement is rigid and the camera pan feels mechanical, lacking the natural micro-movements of real human subjects, while details like hands appear blurred or malformed.“
>
> This comparison suggests that, in mixed scenarios, model judgments are shaped not only by media artifacts but also by cues such as factual plausibility.
>
> ## Q2: Response to the Attribution of OmniChecker’s Gains
>
> The key issue is whether OmniChecker’s gains can be attributed separately.
>
> Fig. 7(a) isolates the **agentic pipeline** by comparing the same frontier MLLM, **Gemini 3 Pro**, under **single-pass multi-task evaluation** and **task-decomposed evaluation within OmniChecker**:
>
> | Model | Setup | Text | Audio | Image | Video | Overall |
> |------|------|------:|------:|------:|------:|------:|
> | Gemini 3 Pro | Single-Pass | 51.60 | 84.42 | 86.83 | 93.09 | 78.99 |
> | Gemini 3 Pro | Agentic Pipeline | 56.20 | 88.54 | 88.68 | 93.42 | 81.71 |
>
> So the **agentic pipeline alone** raises the best overall ceiling from **78.99** to **81.71**; the lower plots in Fig. 7(a) show the same trend for **cross-modal consistency** and **fact verification**.
>
> Fig. 7(c) separately evaluates the **fine-tuned adapter**:
>
> | Model | Setup | Text | Audio | Image | Video | Overall |
> |------|------|------:|------:|------:|------:|------:|
> | Qwen2.5-Omni-3B (LoRA FT) | Specialized Detector | 81.41 | 89.76 | 82.50 | 94.96 | 87.16 |
>
> This detector is a **task-specific OmniChecker module**, not a standalone single-pass model, so its results reflect the contribution of a **specialized adapted component**. Despite its lightweight design, it surpasses **Gemini-3-Pro** on most detection dimensions.
>
> Therefore, we find the **agentic pipeline** improves the upper bound of some frontier MLLMs, while the **fine-tuned adapter** brings additional gains for media-authenticity detection.
>
> ## Q3: Response to Benchmark Quality Assurance
>
> FakeWorld quality is controlled at three levels: **dataset construction**, **template design**, and **explainable annotation**.
>
> **(1) Dataset construction.** FakeWorld is built from a filtered seed corpus with independently verifiable factual content suitable for controlled perturbation (see **Reviewer DkzZ, Q2**). We apply strictly localized factual edits, and ongoing manual refinement further checks their validity. Multimodal consistency labels and generated data consistency are also manually refined (see **Reviewer DkzZ, Q4**).
>
> **(2) Template design.** Our explanation templates are not unconstrained model outputs; they are synthesized from forensic literature, guided by explicit rules, and verified by human experts before large-scale use.
>
> **(3) Explainable annotation.** Initial rationales are generated by a frontier model ensemble to reduce single-model bias, followed by ongoing two-person expert refinement over the full dataset, including correction of hallucinated or unsupported explanations (see **Reviewer RMg8, Q1**).
>
> Overall, quality control is applied throughout the full pipeline, from seed selection and controlled construction to expert-verified templates and dataset-wide human refinement.

---

> > ### Author Rebuttal · Reviewer_2i43 · 2026-04-03
> >
> > Rebuttal addresses my concern. I would like to raise my rating.

---

> > > ### Author Response · Authors · 2026-04-04
> > >
> > > Thank you so much for your thoughtful follow-up and for increasing your score. We sincerely appreciate your careful consideration of our rebuttal, and we are glad that the additional experiments and clarifications are helpful in addressing your concerns.

---

### Official Review · Reviewer_RMg8 · 2026-03-13

**Soundness:** 3
**Presentation:** 4
**Significance:** 3
**Originality:** 3
**Overall Recommendation:** 4
**Confidence:** 4

**Summary:**

This paper presents FakeWorld 1.0, an omni-modal benchmark addressing challenges in high-fidelity deceptive media by combining media authenticity and content veracity into a unified evaluation framework. Using generative models like Sora and Veo 3.1, the authors create a dataset of 3,153 balanced samples across text, audio, image, and video modalities. The work also includes 16,000 structured annotations and introduces OmniCheck, an explainable verification framework that moves from black-box classification to evidence-based diagnostics. FakeWorld 1.0 tests the limitations of current detection methods and provides a foundation for future research in multi-modal forensic analysis and fact-checking.

**Compliance With Llm Reviewing Policy:**

Affirmed.

**Final Justification:**

The paper is well justified, and the proposed method is effective. Regarding the concern I initially raised about human validation, the authors supplemented the rebuttal with relevant experiments and methodological details. Therefore, I retain my initial score and believe that this paper meets the acceptance standard of ICML.

**Key Questions For Authors:**

1. Lack of Human-in-the-Loop Validation for Annotations

2. Absence of Human Baseline Performance

A more detailed explanation can be found in the “weakness” .

**Limitations:**

The paper does not provide its own limitations. I believe that if this work could be further developed into a dynamic and scalable evaluation framework, it would be more in line with the current rapid development of AIGC.

**Strengths And Weaknesses:**

# Strengths

1. Novel Unified Evaluation Dimension: The work introduces a pioneering multi-dimensional benchmark that integrates both media authenticity (AI-generated vs. Natural) and content veracity (Factual vs. Non-factual), bridging the gap between previously isolated forensic and fact-checking tasks.

2. High-Fidelity and Diverse Data Synthesis: By leveraging state-of-the-art generative models and realistic social/news layouts, the dataset provides a high-fidelity simulation of complex, cross-modal deceptive scenarios that closely mimic real-world misinformation environments.

3. Explainable Agentic Verification Framework: The proposed OmniCheck framework shifts from black-box classification to an evidence-based diagnostic approach, utilizing structured rationale-level annotations to provide transparent and interpretable detection results across all modalities.

# Weaknesses

1. Lack of Human-in-the-Loop Validation for Annotations: While the dataset provides over 16,000 rationale-level annotations, these are predominantly generated via structured templates without rigorous human-in-the-loop verification. This absence of manual auditing raises potential concerns regarding the logical consistency and factual accuracy of the automatically generated explanations.

2. Absence of Human Baseline Performance: The evaluation lacks a human study to establish a performance benchmark. Incorporating human accuracy—leveraging innate human intuition for sensory anomalies (e.g., unnatural limb movements or audio-visual dyssync) and news veracity—is essential to quantify the current gap between AI models and human-level discernment.

3. Limitations in Scalability and Adaptability: The framework relies on a relatively static dataset construction, which may limit its extensibility against rapidly evolving generative technologies. Given the swift advancements in AIGC, a more dynamic and scalable detection framework would offer greater practical utility in addressing emerging, unseen deceptive techniques.

---

> ### Author Rebuttal · Authors · 2026-03-31
>
> ## Q1: Response to Concerns Regarding Human-in-the-Loop Validation
>
> We thank the reviewer for the constructive feedback regarding the reliability of our automated annotations. We are currently ensuring the logical rigor of FakeWorld 1.0 through the following multi-stage refinement process:
>
> * **High-Quality Initialization**: All explanations follow structured templates synthesized from existing forensic literature and verified by human experts. Initial rationales were generated by a frontier model ensemble, comprising GPT-5.2, Gemini 3 Pro, and Qwen3-VL-235B-thinking, following these expert-vetted templates to minimize individual model bias.
>
> * **Ongoing Mandatory Expert Refinement**: We have implemented an ongoing two-person manual audit process (expert revision and verification) that is being applied to all 3,153 instances. Our team strictly judges the validity of existing explanatory annotations, deleting or modifying problematic explanations where model hallucinations occurred, and adding explicit, evidence-based rationales. This workflow also includes manually revising consistency ground truths, as addressed in our response to **Reviewer DkzZ (Q4)**.
>
> * **Quantitative Progress**: To date, our team of six authors has completed manual refinement for **over 1,200 explanatory annotations**. Within this audited subset, the proportion of annotations containing **clear hallucinations or incorrect explanations is below 1%**, indicating that these issues are limited and do not materially affect our main conclusions. We will continue this screening and refinement process for the remaining annotations.
>
> ## Q2: Response to Absence of Human Baseline Performance
>
> We agree that establishing a human baseline is essential for quantifying the performance gap between state-of-the-art MLLMs and human-level discernment. Human intuition for sensory anomalies and linguistic nuances provides a critical benchmark for evaluating fake content detection. To address this, we conducted a human study on a representative subset of the FakeWorld benchmark. Results compared against frontier models are summarized below:
>
> | Model / Baseline | Fact Check | AI Text | AI Image | AI Audio | AI Video | T-I Consist. | T-A Consist. | T-V Consist. |
> | :--- | :---: | :---: | :---: | :---: | :---: | :---: | :---: | :---: |
> | GPT-5.2 | 81.86 | **68.47** | 53.22 | - | 73.73 | 65.56 | - | **71.28** |
> | Gemini 3 Pro | **87.65** | 51.60 | 86.83 | 84.42 | 93.09 | 66.40 | **89.74** | 62.04 |
> | **Human Baseline** | 82.24 | 65.26 | **88.80** | **93.01** | **96.12** | **70.32** | 85.44 | 53.79 |
>
> In our analysis, fact-checking strongly relies on access to external knowledge and encyclopedic information, which gives models an advantage over humans. For AI-generated content detection, humans generally perform well, but it remains challenging under our News- and encyclopedia-oriented AI text scenarios. Regarding consistency, humans lag behind models, particularly in video modalities. This occurs because, although each instance originates from a single semantic base, modifications of factual details in generated multimedia content create subtle differences in semantics and fine-grained details. Once these differences explicitly appear in the content, they constitute semantic inconsistencies. In our analysis of human consistency errors, we found that humans mislabeled inconsistent instances as consistent **up to 85% of the time**, highlighting this challenge.  For example, in a real dataset **Text-Video Consistency Cue Explanation** part:
>
> > Entities (People): There is a distinct mix of personnel present..., and uniforms (some brown, some dark blue) with the text **"RESCATE" (Spanish for "rescue")** readable on their gear. They are working...
>
> This type of fine-grained semantic detail and the associated knowledge requirement illustrate why models, with broader knowledge and more precise observation, are better equipped to detect inconsistencies.
>
> ## Q3: Response to the Explainable Agentic Verification Framework
>
> We thank the reviewer for the comment on scalability and adaptability.
>
> FakeWorld emphasizes a **joint media and semantic axis**, integrating text, audio, image, and video with factual and cross-modal consistency. This structured framework naturally supports **next-generation, harder-to-detect instances** as generative models advance.
>
> OmniCheck is modular and decoupled: task decomposition, media/content verification, and explainable verdict generation allowing seamless integration of **stronger MLLMs or domain-specific models**. Explainable templates can be updated with **new detection strategies**, keeping the verification pipeline and interpretability evolving.
>
> We will further explore deeply integrated dataset construction, e.g., creating highly deceptive samples where video captions and scene semantics closely align with the textual semantic root.

---

> > ### Author Rebuttal · Reviewer_RMg8 · 2026-04-04
> >
> > As the discussion stage is not yet over, I will refer to the opinions of other reviewers to give the final score.

---

> > > ### Author Response · Authors · 2026-04-04
> > >
> > > Thank you very much for the acknowledgement. We sincerely appreciate that our concerns have been fully addressed in your view, and we also appreciate your careful consideration of the broader reviewer discussion before determining the final score.

---

### Decision · Program_Chairs · 2026-04-30

**Decision:**

Accept (regular)

**Comment:**

Summary:
This paper introduces FakeWorld 1.0, a new benchmark that combines 4 modalities (text, audio, image, and video synthesis) to address fake media and AI-generated content, along with OmniCheck, a unified evaluation and verification framework to demonstrate its usefulness. Using Sora and Veo 3.1, the authors created over 3k+ samples across all 4 modalities and 16k+ structure annotations.

Justifications:
FakeWorld 1.0 is a novel benchmark that combines 4 modalities together to make the fake media content more realistic and challenging. It bridges the gap between previous work in CV and NLP communities on fake visual and textual content. The created dataset and the OmniCheck have potential for researchers and AI practitioners. The authors' rebuttal fully addressed the concerns from reviewers RMg8 and 2i43, as well as the main concerns from reviewer 8jxn.